# Single-unit activations confer inductive biases for emergent circuit solutions to cognitive tasks

Pavel Tolmachev ⓘ & Tatiana A. Engel ⓘ ✉

Trained recurrent neural networks (RNNs) have become the leading framework for modelling neural dynamics in the brain, owing to their capacity to mimic how population-level computations arise from interactions among many units with heterogeneous responses. RNN units are commonly modelled using various nonlinear activation functions, assuming these architectural differences do not affect emerging task solutions. Here, contrary to this view, we show that single-unit activation functions confer inductive biases that influence the geometry of neural population trajectories, single-unit selectivity and fixed-point configurations. Using a model distillation approach, we find that differences in neural representations and dynamics reflect qualitatively distinct circuit solutions to cognitive tasks emerging in RNNs with different activation functions, leading to disparate generalization behaviour on out-of-distribution inputs. Our results show that seemingly minor architectural differences provide strong inductive biases for task solutions, raising a question about which RNN architectures better align with mechanisms of task execution in biological networks.

Recurrent neural networks (RNNs) offer a versatile framework for modelling mechanisms of cognitive computations in the brain[1–6]. Similar to biological neural circuits, RNNs consist of many interconnected units with nonlinear activation function, mimicking the nonlinear input-to-output transformation of individual neurons. Whether through training to perform behavioural tasks[4,7,8] or by directly fitting recorded neural activity[9–11], RNN units develop heterogeneous responses similar to the mixed selectivity observed in brain recordings[2]. Thus, RNNs serve as computationally tractable models that capture key features of biological neural networks. Analysing task solutions that emerge in RNNs through training provides hypotheses for how biological networks may execute similar tasks[1,8,10,12].

Continuous-time RNNs are universal approximators of any dynamical system. RNNs can approximate the desired dynamics with arbitrary precision in a subset of output units, provided a sufficient number of hidden units in the network[13,14]. Although the proof of this result holds for smooth and bounded sigmoid-like activation functions, the empirical evidence suggests that RNNs with rectified linear (ReLU) activation may approximate complex dynamics as well[3,4]. Accordingly, it is commonly assumed that a specific choice of activation function is inconsequential to the mechanisms that emerge in RNNs, as long as the networks are adequately trained to perform the task. Supporting this assumption, a comprehensive study of RNNs with different architectures found that, despite some differences in the geometry of neural dynamics, they use similar computational scaffolds, as characterized by the topological structure of fixed points[15]. Consequently, many studies employed a variety of activation functions to model biological networks, including sigmoid[16,17], ReLU[3,4,12,18–20] and hyperbolic tangent (tanh)[7,8,21–25]. However, whether these architectural choices are truly inconsequential to the circuit mechanisms emerging in RNNs through training has not been systematically tested.

We hypothesized that seemingly minute differences in the geometry of neural representations across RNN architectures[15,26] may reflect deeper distinctions in the underlying circuit mechanisms driving

Princeton Neuroscience Institute, Princeton University, Princeton, NJ, USA. ✉e-mail: tatiana.engel@princeton.edu

behaviour. To test this hypothesis, we analysed RNNs with six architectures trained on a range of tasks. We used three common activation functions (ReLU, sigmoid and tanh) and, for each, trained RNNs with and without Dale's law constraint on the connectivity (restricting units to be either excitatory or inhibitory)[3,27], a fundamental feature of cortical circuits. Neural representations and dynamics differed across RNNs with varying activation functions, with tanh networks diverging the most from both sigmoid and ReLU RNNs. Using a model distillation approach[12], we uncovered that these differences arose from distinct circuit solutions used by the RNNs to solve the same task. Moreover, these circuit solutions made disparate predictions for how RNNs respond to out-of-distribution inputs, which were confirmed through simulations. Our findings imply that conclusions about mechanisms of task execution derived from reverse-engineering RNNs may depend on subtle architectural differences, emphasizing the need to identify architectures with inductive biases that most closely align with biological data.

## Results

How profound are differences across networks trained with varying architectures, such as different activation functions and connectivity constraints? To answer this question, we trained RNNs with various architectures on a range of tasks. We used three activation functions (ReLU, sigmoid and tanh) and, for each, trained 100 networks both with and without Dale's law connectivity constraint (Dale, no Dale), resulting in six distinct architectures. All RNNs were trained on the same task inputs and outputs to a similar performance level (Extended Data Fig. 1 and Extended Data Table 1). Our analysis focused on the 50 top-performing networks from each architecture, and the remaining 50 networks yielded similar results (Extended Data Fig. 2).

Across all RNNs, we compared neural representations through population trajectories and single-unit selectivity and dynamical mechanisms characterized by fixed-point and trajectory endpoint configurations. We further extracted circuit mechanisms driving task behaviour in these RNNs and tested their generalization performance on out-of-distribution inputs. We first present our findings for a context-dependent decision-making (CDDM) task (Fig. 1 and 'CDDM task' section in Methods) and then show that these observations generalize to other tasks.

### Differences in representations across RNN architectures

We compared representations in trained RNNs by analysing the geometry of trajectories in the population state space and single-unit tuning in the selectivity space—two complementary perspectives on neural responses[28-31]. Responses of a single RNN with $N$ units across $K$ trials, each with $T$ time steps, form a matrix of shape ($N$, $T \times K$). Each column of this matrix defines a point in the $N$-dimensional population state space. Each row of the same matrix represents the tuning profile of a single unit in the selectivity space. Accordingly, we reduce dimensionality with principal component (PC) analysis applied to either the columns or rows of the neural response matrix. For population trajectories, we reduce the first dimension from $N$ to $n_{PC}$ yielding a matrix of shape ($n_{PC}$, $T \times K$) containing a set of projected trajectories ('Analysis of population trajectories' section in Methods). For single-unit selectivity, we reduce the second dimension from $T \times K$ to $n_{PC}$, resulting in a matrix of shape ($N$, $n_{PC}$) containing projected selectivity profiles of RNN units ('Analysis of single-unit selectivity' section in Methods).

For initial assessment, we visualized population trajectories of example RNNs by projecting them onto the first two PCs. The trajectories of ReLU and sigmoid RNNs were visually distinct from those of tanh networks (Fig. 2a). ReLU and sigmoid RNNs typically form symmetric, butterfly-shaped trajectory sets: the trajectories remain near the origin during the presentation of the context cue at the trial start and gradually separate later in the trial when sensory inputs are introduced. By contrast, the trajectories of tanh RNNs diverge immediately at trial onset, driven solely by context inputs, and further separate based on

sensory inputs later in the trial, forming two sheets orthogonal to the context axis. Dale's constraint did not affect the geometry of population trajectories in tanh RNNs. In ReLU and sigmoid RNNs, Dale's constraint produced more structured representations, with trajectories clustering by context and choice, whereas the trajectories varied more continuously in unconstrained networks.

To systematically quantify these differences across RNNs, we embedded individual trajectory sets into a shared two-dimensional space, where each point represents a single RNN and distances between points reflect dissimilarity between trajectory sets (Fig. 2b and 'Analysis of population trajectories' section in Methods). We defined dissimilarity as the mean squared error (m.s.e.) between projected trajectories ($n_{PC} = 10$) of two RNNs after optimal alignment using orthogonal Procrustes. Using these pairwise dissimilarities, we embedded all RNNs into a two-dimensional space using multidimensional scaling (MDS)[15], which aims to minimally distort all pairwise distances. This analysis confirmed our initial observations: the networks with different architectures formed distinct clusters in the embedding space (Fig. 2b). Tanh RNNs, with or without Dale's constraint, were clearly separated from ReLU and sigmoid RNNs. The clusters formed by ReLU and sigmoid RNNs without Dale's connectivity constraint show higher spread than their constrained counterparts, indicating larger heterogeneity in population trajectories across networks.

We further examined single-unit selectivity configurations, which also differed across RNNs with different architectures. Visualizing single-unit selectivity in example RNNs reveals striking differences between ReLU and sigmoid versus tanh networks (Fig. 2c). ReLU and sigmoid RNNs produce a cross-shaped pattern with continuously populated arms extending outward, whereas tanh RNNs display a large central cluster with a few distant, outlying units. We computed pairwise distances between the single-unit selectivity configurations across all RNNs and embedded them into two-dimensional space using MDS. A pairwise distance was computed as m.s.e. between selectivity configurations of two RNNs after aligning them using an iterative closest point (ICP) registration algorithm, which permits one-to-many unit matching ('ICP registration' section in Methods). This analysis confirmed that tanh RNNs are distinct from ReLU and sigmoid networks (Fig. 2d). Furthermore, the embedding revealed that tanh RNNs with and without Dale's constraint form a single cluster. By contrast, ReLU and sigmoid RNNs with and without Dale's constraint form clearly separable clusters.

Thus, the analyses of population trajectories and single-unit selectivity revealed that neural representations in tanh networks are distinct from those in ReLU and sigmoid RNNs. These differences were also evident in trained networks with shuffled connectivity (Extended Data Fig. 3) and even in randomly initialized networks (Extended Data Fig. 4), and were further amplified through training. In addition, Dale's connectivity constraint does not affect neural representations in tanh networks, contrasting with ReLU and sigmoid RNNs.

### Differences in dynamics across RNN architectures

Having observed distinct neural representations across RNN architectures, we next asked whether these differences reflect distinct dynamical mechanisms for solving the task. We characterized dynamical mechanisms by analysing the fixed-point configurations in RNNs with various architectures[15]. Fixed points are the states of a dynamical system where the flow field vanishes under constant input, that is, once the RNN state reaches a fixed point, it remains unchanged unless perturbed. Fixed-point configurations provide a computationally tractable description of task-relevant dynamics in RNNs[8,32].

In each RNN, we computed the fixed points for each combination of task inputs ('Fixed-point finder' section in Methods). Specifically, for the CDDM task, we computed fixed points with both the context and sensory inputs held constant for a total of 50 distinct input

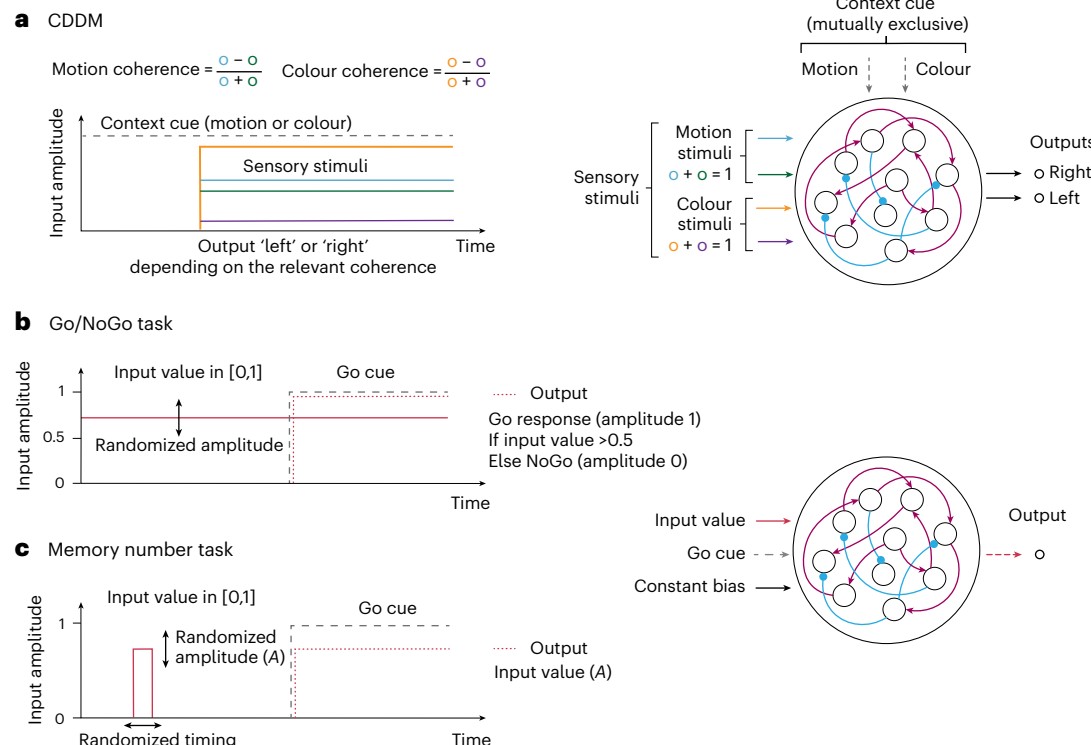

**Fig. 1 | Tasks used for RNN training. a**, Right: CDDM task. RNN receives two modalities of sensory inputs, termed 'motion' and 'colour'. For each modality, two positively constrained input channels provide momentary evidence for left and right choice, respectively. The difference between the mean right and left input defines stimulus coherence, with values ranging from −1 to 1. Two context channels supply the cued context input, with only one context channel active on each trial indicating the relevant sensory modality. The network is required to output 'left' or 'right' decision on the corresponding output channel based on the signed coherence of the relevant sensory input. Left: time-course of the CDDM task. The 'context cue' is present throughout the trial, indicating either 'motion' or 'colour' context. Following the context cue, four channels convey sensory stimuli (two sensory modalities with two channels per modality). The network is required to produce an output as soon as the sensory inputs are supplied. **b**, Go/NoGo task. A single input value chosen from (0, 1) range is presented throughout the trial. Upon presentation of the 'Go cue', the network is required to output 'Go' response with amplitude 1 if the input value is above the 0.5 threshold and produce a 'No Go' response with amplitude 0 if the input value is below 0.5. **c**, The memory number task. An input value with randomized amplitude $A$ is briefly presented at the beginning of the trial within a randomized time window. Upon receiving a 'Go cue', the RNN is required to output the same value $A$. Right: Go/NoGo and memory number tasks have shared input and output structure.

combinations (five relevant and five irrelevant coherences in two contexts). We aggregated the fixed points from all inputs and assessed their stability, categorizing each fixed point as either stable or unstable. We then compared the resulting fixed-point configurations across RNNs with different architectures.

First, we visualized the fixed-point configurations of example RNNs by projecting them onto the first two PCs (Fig. 3a). ReLU and sigmoid RNNs showed similar fixed-point configurations. Their fixed points were clearly separated along the second PC according to the context cue. Within each context, the stable fixed points clustered at the extremes of the first PC, corresponding to left and right choices, with the unstable fixed points located in between. The stable fixed points in ReLU and sigmoid RNNs formed elongated clusters, indicating that irrelevant stimulus is still represented, albeit to a limited degree. By contrast, tanh RNNs displayed sheet-like fixed-point configurations, with irrelevant information being less suppressed, as evidenced by the nearly uniform distribution of fixed points across each sheet. While the fixed-point configurations of tanh networks were unaffected by Dale's constraint, ReLU and sigmoid RNNs showed less variability in the fixed-point configurations under this constraint compared with when it was absent.

To quantify these differences across all RNNs, we embedded their fixed-point configurations into a two-dimensional space using MDS (Fig. 3b and 'Analysis of fixed points' section in Methods). We computed pairwise distances between RNNs as an m.s.e. between their fixed points aligned with a custom registration algorithm, which accounted for the fixed-point type (stable or unstable) for each input. The resulting MDS embedding confirmed our initial observations: while all architectures were separable, the tanh RNNs clustered further away from the ReLU and sigmoid networks. We further verified these results by analysing configurations of trajectory endpoints (network state at the last time step of a trial), which tend to converge towards stable fixed points. As expected, the trajectory endpoints mirrored the fixed-point configurations (Fig. 3c), and their MDS embedding further reinforced that tanh RNNs are distinct from both ReLU and sigmoid networks (Fig. 3d). By contrast, different architectures were largely indistinguishable in the MDS embeddings of fixed-point or trajectory endpoint configurations in networks with shuffled (Extended Data Fig. 3) or randomly initialized connectivity (Extended Data Fig. 4 and Extended Data Fig. 5), because fixed points merely reflect the shared input structure in random networks.

## RNNs with varying architectures rely on different circuit mechanisms

Given the differences in neural representations and dynamics across RNN architectures, we asked whether these variations reflect distinctness of circuit solutions discovered by each RNN class for the same task. To identify the circuit mechanism used by each RNN to solve the CDDM task, we fitted its neural responses and task behaviour with a latent circuit model[12] ('Latent circuit inference' section in Methods). Specifically, we fit RNN responses as a linear embedding of dynamics generated by a low-dimensional RNN—the latent circuit—which has

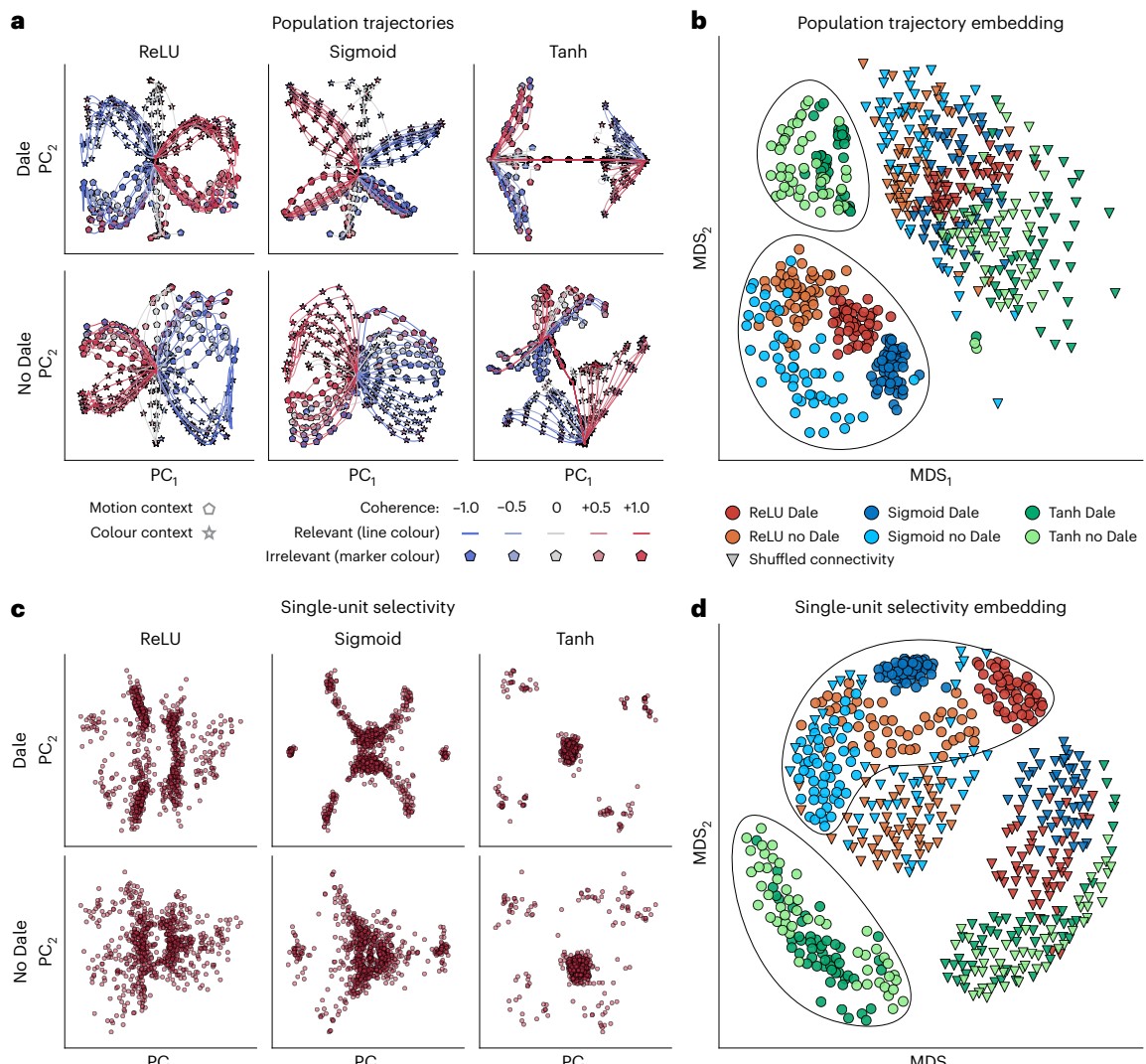

**Fig. 2 | Comparison of population trajectories and single-unit selectivity across six RNN architectures trained on the CDDM task. a**, Population trajectories visualized by projecting onto the first two PCs in example RNNs with different activation functions (columns) and connectivity constraints (rows). **b**, An MDS embedding of population trajectory sets across all RNNs. Each point represents the set of trajectories from a single RNN. ReLU and sigmoid networks form clusters distinct from tanh networks. The triangles represent trajectories of the same RNNs with shuffled connectivity matrices, used as a control. **c**, Single-unit selectivity visualized by projecting onto the first two PCs in RNNs with different activation functions (columns) and connectivity constraints (rows). Each point represents one unit. Each plot aggregates units from the top 30 RNNs, showing only units with activity levels above the 50th percentile. **d**, An MDS embedding of single-unit selectivity configurations across RNNs. Each point represents one RNN. RNNs with each architecture form distinct clusters, with the tanh RNN cluster positioned further away from the others. In **b** and **d**, each RNN architecture is represented by the top 50 RNNs (600 RNNs in total, including controls).

the same activation function and is also required to reproduce task outputs. Thus, the latent circuit model infers a low-dimensional circuit mechanism generating task-relevant dynamics in the RNN. We inferred latent circuits for the ten top-performing RNNs from each architecture. All latent circuits produced accurate fits while also successfully solving the CDDM task (Table 1).

The inferred latent circuit connectivity revealed that ReLU and sigmoid RNNs rely on a mechanism distinct from that of tanh RNNs to select relevant stimuli in the CDDM task. In ReLU and sigmoid RNNs, context nodes inhibit sensory nodes representing irrelevant stimuli in each context (Fig. 4a, for example, motion context node inhibits sensory nodes representing colour). Since the activity of irrelevant sensory nodes is suppressed, only the relevant nodes drive the choice output[12]. By contrast, tanh RNNs use a qualitatively different circuit solution (Fig. 4c). The active context node drives the nodes representing relevant and irrelevant stimuli to the opposite saturation regions

of the tanh activation function. Before stimulus onset, the RNN output is precisely zero due to a stalemate between the negatively saturated relevant nodes and positively saturated irrelevant nodes. A positive stimulus drives the relevant nodes to the steep region of the tanh activation function affecting the output, while stimulus input does not change the activity of the positively saturated irrelevant nodes and, hence, has no effect on choice.

These different circuit mechanisms make distinct predictions for how networks respond to out-of-distribution inputs. The tanh circuit predicts that increasing the amplitude of the irrelevant stimulus beyond the range used during training will not affect the output, because this input will push the irrelevant nodes further in the positive tanh saturation region without changing their activity. By contrast, the ReLU circuit predicts that a sufficiently strong irrelevant input will overcome the inhibition of the irrelevant nodes by the context nodes, allowing the irrelevant stimulus to bias the output. These predictions

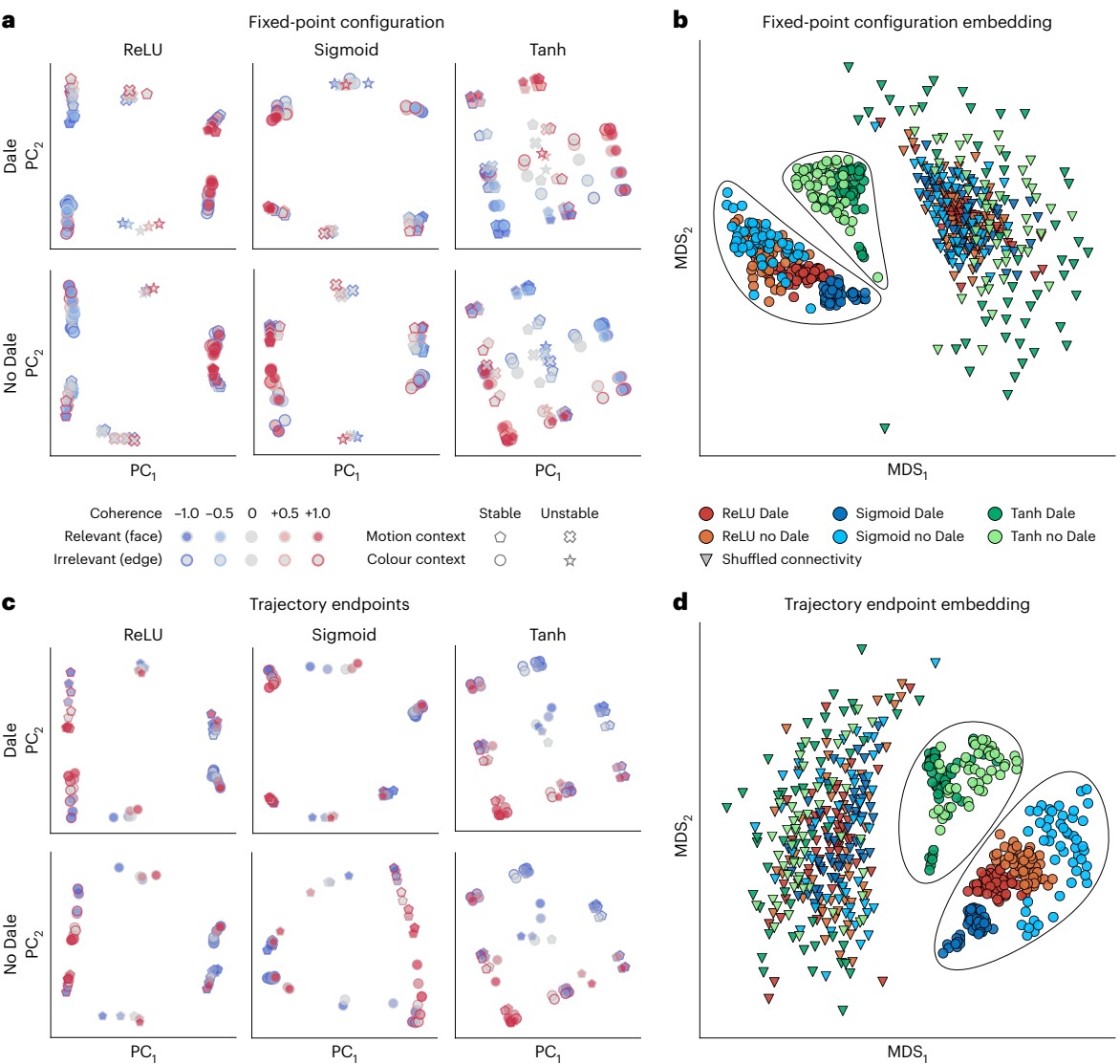

**Fig. 3 | Comparison of fixed-point and trajectory endpoint configurations across six RNN architectures trained on the CDDM task. a**, Fixed-point configurations visualized by projecting onto the first two PCs in example RNNs with different activation functions (columns) and connectivity constraints (rows). We compute stable and unstable fixed points for each of 50 possible input combinations to the RNN (two context cues, five relevant and five irrelevant coherences). **b**, An MDS embedding of fixed-point configurations across top 50 RNNs from each architecture. Each point represents the fixed-

point configuration of one RNN. ReLU and sigmoid networks form clusters distinct from tanh networks. The triangles represent fixed points of the same RNNs with shuffled connectivity matrices, used as a control. **c**, Trajectory endpoint configurations visualized by projecting onto the first two PCs mirror the configurations of stable fixed points in example RNNs. **d**, An MDS embedding of trajectory endpoint configurations across all RNNs. In **b** and **d**, each RNN architecture is represented by the top 50 RNNs (600 RNNs in total, including controls).

were clearly borne out in simulations. When exposed to irrelevant inputs with amplitudes greater than during training, ReLU and sigmoid RNNs became sensitive to these irrelevant stimuli, evident as a rotation of the decision boundary in the psychometric function (Fig. 4b). By contrast, strong irrelevant stimuli with amplitudes beyond the training range had no effect on the psychometric function of tanh networks (Fig. 4d). These results demonstrate that circuit mechanisms define how networks generalize to out-of-distribution inputs and that different RNN architectures carry inductive biases favouring different circuit mechanisms.

Together, our results show that differences in population trajectories, single-unit selectivity and fixed-point configurations can indicate distinct circuit solutions discovered by RNNs for the same task. Moreover, RNN architectures impose inductive biases that favour specific circuit solutions, highlighting the importance of the architectural choice in modelling biological data.

## Differences in RNN architectures across tasks

Are the differences in neural representations and dynamics across RNN architectures specific to the CDDM task, or do they manifest in other tasks as well? To answer this question, we trained RNNs with all six architectures to perform the Go/NoGo and memory number tasks (Fig. 1b,c and 'Go/NoGo and memory number tasks' section in Methods). We compared neural representations and dynamics in these RNNs using population trajectories, single-unit selectivity and fixed-point configurations.

In both tasks, example ReLU and sigmoid RNNs showed qualitatively similar projected trajectories, which differed from the trajectories of tanh networks (Fig. 5a,b). In ReLU and sigmoid RNNs, low-amplitude inputs generated tightly compressed trajectories, while high-amplitude inputs drove large excursions through the state space. In tanh RNNs, by contrast, the extent of trajectories was more similar between high- and low-amplitude inputs.

**Table 1 | Latent circuit fit accuracy**

| | ReLU | | Sigmoid | | Tanh | |
|---|---|---|---|---|---|---|
| | **Dale** | **No Dale** | **Dale** | **No Dale** | **Dale** | **No Dale** |
| $R^2$, RNN dynamics | 81% ± 1% | 71% ± 3% | 81% ± 2% | 75% ± 2% | 92% ± 1% | 89% ± 1% |
| $R^2$, target behaviour | 96% ± 1% | 95% ± 1% | 96% ± 1% | 95% ± 1% | 97% ± 1% | 97% ± 1% |

Coefficient of determination $R^2$ for the trajectories (upper row) and behavioural output (lower row) of the fitted latent circuit compared with the target RNN. The data are mean±s.d. across the best-fitting latent circuits for each of the top ten RNNs trained on the CDDM task within each architectural class.

MDS embeddings of each metric across all RNNs revealed that ReLU, sigmoid and tanh networks formed distinct clusters for both tasks (Fig. 5c,d). Although ReLU and sigmoid networks clustered separately, they were consistently closer to each other than to tanh networks, confirming the distinctive behaviour of tanh RNNs across all three tasks. Within each activation function, networks with and without Dale's constraint formed partially overlapping subclusters, which were closer to each other than to any subclusters corresponding to other activation functions. These results indicate that while Dale's constraint influences representations and dynamics, the activation function has a substantially greater effect. These findings reinforce that the choice of activation function impacts emergent task solutions, with tanh networks standing out as the most distinct from their ReLU and sigmoid counterparts.

## Discussion

We show that RNN architectures confer inductive biases that influence neural population dynamics, single-unit selectivity and circuit mechanisms emerging through training on cognitive tasks. Different circuit mechanisms manifest in diverging behaviour on out-of-distribution inputs, demonstrating that these differences reflect fundamentally distinct task solutions rather than mere trivial variations. Task-optimized RNNs are widely used to generate hypotheses for how the brain may solve cognitive tasks, yet the choice of activation function is often assumed inconsequential for the resulting mechanisms, reflected in a variety of activation functions used across studies[3,4,7,8,12,16–25]. Our findings indicate that different architectures can yield disparate circuit solutions, which may vary in their alignment with circuit mechanisms in the brain.

Our findings have broader implications for methods that directly optimize RNNs to reproduce neural recording data, such that each RNN unit tracks the activity of one experimental neuron[9–11,33]. Some architectures may be more amenable than others to replicating neural recordings. While RNNs can approximate any dynamics given a sufficient number of units, the number of units required to achieve a given level of accuracy may depend on how well the architecture aligns with the structure and constraints of biological circuits. Architectures more closely aligned with biological circuits may require fewer units to fit the neural dynamics accurately. Moreover, it remains an open question whether two architectures that can equally well fit the same neural responses converge on the same circuit solution. Therefore, the choice of RNN architecture cannot be ignored when modelling biological systems, as these choices may bias the inferred solutions and their relevance to neural processes.

We find that the geometric arrangement of fixed points differs consistently across RNN architectures for multiple tasks, whereas previous work has shown that the topology of fixed-point configurations is universal across architectures for certain tasks[15]. The topology can be characterized by a directed graph, where the vertices correspond to fixed points and the edge weights indicate the probability of trajectories diverging from one fixed point to another[15]. While for some tasks, the fixed-point topology was universal across architectures, mirroring the underlying computational scaffold, it differed between ReLU and tanh networks in a context-dependent integration task[15]. Since topology disregards the fixed points' locations in the state space, a universal

topology can be consistent with varying geometries of fixed points. Our results indicate that different fixed-point geometries correspond to distinct circuit solutions to the task, producing divergent behaviour on out-of-distribution inputs. Therefore, topological universality is not sufficient for characterizing task solutions and does not imply equivalence of different architectures for modelling the brain.

We show that RNNs with different architectures exhibit distinct configurations in the selectivity space, suggesting that single units assume different functional roles across architectures. Consistent with this result, previous studies found that ReLU RNNs trained to perform many cognitive tasks develop functional clusters of units specialized for subsets of tasks, whereas tanh RNNs showed broader selectivity with noticeably fewer clusters[8,34]. Ablations of these functional unit clusters produced task-specific behavioural deficits[34], echoing our conclusion that neural representations are indicative of the underlying causal mechanisms. Moreover, single-unit selectivity resembling experimentally observed grid cells in the entorhinal cortex emerged in task-optimized RNNs with ReLU activation function but not in tanh RNNs[35]. Thus, the choice of activation function can affect the alignment of the emergent neural representations with biological data[35,36].

Whether activation functions produce similar or distinct circuit solutions may depend on the specific computation. In the CDDM task, ReLU and sigmoid networks produced solutions that were more similar to each other than to those of tanh RNNs. In the Go/NoGo and memory number tasks, sigmoid networks often produced solutions that were as distinct from ReLU networks as from tanh RNNs. Thus, similarities between activation functions are task-dependent rather than absolute. Yet, it is difficult to envision a task in which the solution would differ between ReLU and softplus activation $\frac{1}{\beta}\log(1 + \exp(\beta\mathbf{x}))$ with $\beta \gg 1$, which closely approximates ReLU. Conversely, decreasing $\beta$ makes softplus function more linear, and for sufficiently small $\beta$, the network may gradually—or even abruptly—lose its ability to solve nonlinear tasks altogether. Thus, small changes in the activation function can, in principle, lead to discontinuous shifts in the resulting solution. Optogenetic stimulation experiments in vivo reveal that cortical neurons exhibit a supralinear-to-linear input–output function, intermediate between soft ReLU and sigmoid[37]. Adding such precise biological constraints may steer artificial models toward better alignment with biological circuits, enhancing their ability to generate relevant hypotheses.

While we trained six architectures on three tasks to achieve comparable performance, some architectures may be better suited for specific tasks than others[38]. Although no formal framework exists to predict the optimal activation function for a given task, certain activation properties can enable more efficient solutions to specific tasks. As an example, consider the 3-bit flip-flop task, in which three outputs store independent memory bits, each set by transient +1 or −1 pulses from corresponding inputs and held until the next pulse[32]. Tanh RNNs can solve this task with just three units, as a single tanh unit can sustain bistable activity—positive or negative—through strong self-excitation. By contrast, a single ReLU unit cannot produce bistability, requiring a larger recurrent circuit to maintain and flip each bit in ReLU RNNs. A formal theory identifying which activation functions best suit specific task demands remains an important goal for future research.

In the brain, single-unit specialization may be supported by a diversity of cell types, each potentially fine-tuned for distinct

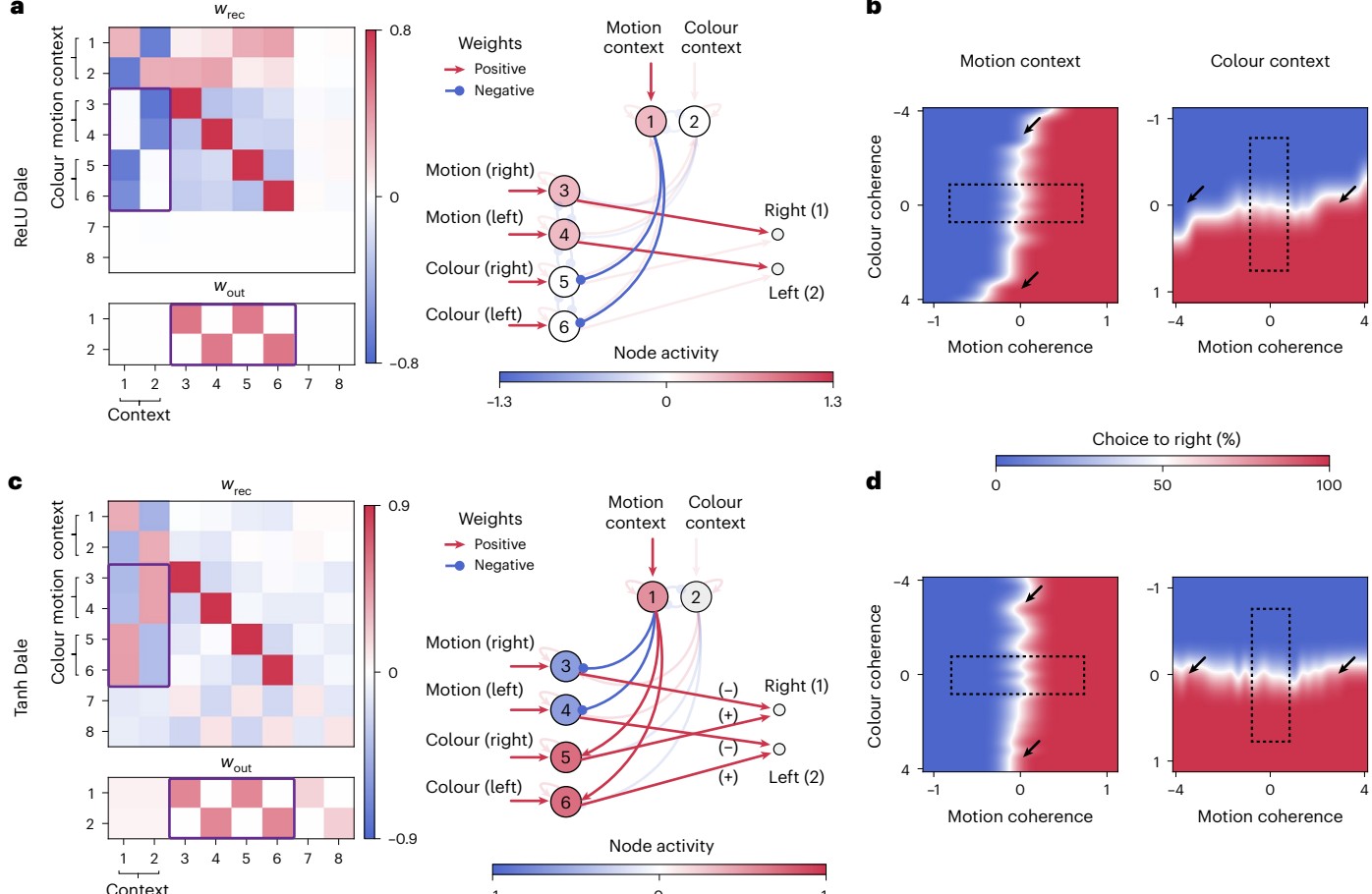

**Fig. 4 | Distinct circuit solutions for CDDM task in ReLU versus tanh RNNs.**
**a**, Left: latent circuit connectivity inferred from responses of a ReLU RNN trained on the CDDM task, including recurrent ($w_{rec}$) and output ($w_{out}$) connectivity matrices. Right: a simplified circuit diagram highlights only the key nodes and connections for clarity, with the filled colour representing the activity of each node on a motion context trial with both context cue and zero-coherence stimuli present. The latent circuit reveals a mechanism for selecting relevant stimuli based on inhibition of nodes representing irrelevant stimuli. The sensory nodes representing motion and colour stimuli project to the corresponding outputs (purple rectangle in $w_{out}$, red arrows in the circuit diagram). Inhibitory connections from the context to sensory nodes (purple rectangle in $w_{rec}$, blue arrows in the circuit diagram) suppress the irrelevant stimulus representations in each context. **b**, The psychometric functions of the ReLU RNN for stimuli extending beyond the range used during training (rectangle indicates the stimulus range used for training). The network becomes sensitive to irrelevant stimuli with increased amplitude, evident as a rotation of the decision boundary

(arrows). Sigmoid RNNs showed qualitatively similar latent circuit mechanism and out-of-distribution behaviour (data not shown). **c**, Same as **a** for latent circuit connectivity inferred from responses of a tanh RNN trained on the CDDM task. The latent connectivity reveals a mechanism for selecting relevant stimuli based on saturation of nodes representing irrelevant stimuli. Context nodes drive the nodes receiving irrelevant stimuli into the positive saturation region of the tanh activation function, while pushing the nodes receiving relevant stimuli into the negative saturation region (purple rectangle in $w_{rec}$). Before stimulus presentation, the negative activity of relevant nodes and positive activity of irrelevant nodes cancel each other at the output (purple rectangle in $w_{out}$, red arrows in the circuit diagram). Relevant stimuli drive the relevant nodes into the steep region of the tanh activation function, allowing them to affect the output. At the same time, irrelevant stimuli drive the irrelevant nodes further into the saturation region, where their activity remains unchanged, thus having no effect on the output. **d**, Same as **b** for the tanh RNN. Strong irrelevant stimuli do not affect the network's choice (arrows).

computations[39–41]. Our Dale-constrained ReLU and sigmoid networks incorporate two basic cell types: excitatory and inhibitory. Neural representations in RNNs with these two cell types differed from their counterparts without Dale's constraint, suggesting that the existence of multiple cell types also affects task solutions emerging in RNNs. Moreover, incorporating multiple cell types with diverse activation functions improved the image classification accuracy of convolutional neural networks compared with conventional homogeneous architecture[42,43]. Thus, equipping RNNs with multiple unit types, featuring different activation functions and connectivity constraints that correspond to biological cell types, may yield closer alignment with biological circuits and higher computational efficiency.

Some activation functions may be effective for solving tasks in artificial networks but not correspond to feasible single-unit dynamics in biological networks. For example, biological neurons cannot produce

negative firing rates, making tanh units biologically implausible. Tanh units reverse the sign of their synaptic effect depending on their activity state, a feature that can be useful for solving certain tasks but not observed in biological neurons. In tanh networks, Dale's constraint loses its biological relevance in defining excitatory and inhibitory cell types. Consistently, this constraint did not influence representations in our tanh RNNs. In addition, network dynamics were less aligned with the output subspace[44] in tanh RNNs than in ReLU or sigmoid networks ('Alignment of RNN dynamics with the output subspace' section in Methods and Extended Data Table 2). Thus, tanh RNNs consistently diverged from the more biologically plausible ReLU and sigmoid RNNs across all metrics and tasks we examined.

Moreover, tanh RNNs produced a circuit mechanism for CDDM that blocks irrelevant stimuli even with arbitrarily large amplitudes. By contrast, ReLU and sigmoid networks become increasingly sensitive

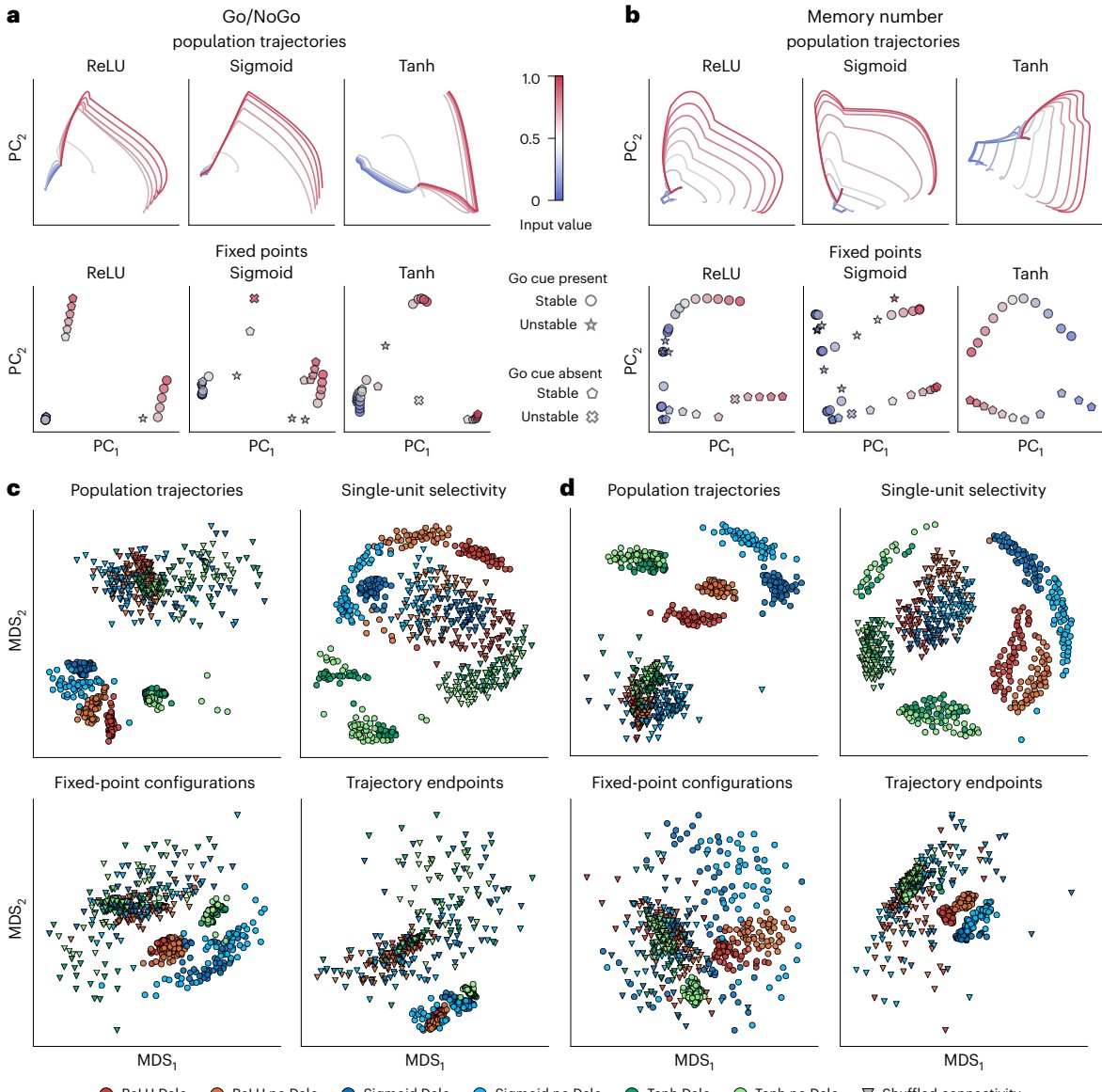

**Fig. 5 | Comparison of population trajectories, single-unit selectivity, fixed-points and trajectory endpoint configurations across RNN architectures for the Go/NoGo and memory number tasks. a**, The population trajectories (upper row) and fixed-point configurations (lower row) projected onto the first two PCs for the Go/NoGo task, for example, ReLU, sigmoid and tanh RNNs with the Dale connectivity constraint. **b**, Same as **a** for the memory number task. **c**, MDS embeddings of population trajectories, single-unit selectivity, fixed points and trajectory endpoints across RNNs trained on the Go/NoGo task. Each point in the embedding space represents a single RNN. The triangles represent data from the same RNNs with shuffled connectivity matrices, used as a control. In all embedding spaces, tanh networks form distinctly separated clusters, while ReLU and sigmoid networks, although distinct, are typically closer to each other. **d**, Same as **c** for memory number task. In **c** and **d**, each RNN architecture is represented by the top 50 RNNs (600 RNNs in total, including controls).

to stronger irrelevant stimuli. In practice, human behaviour is often influenced by strong irrelevant stimuli, as demonstrated by the Stroop effect, where error rates increase when responding to incongruent stimuli[45]. Thus, although tanh RNNs discover a more robust solution, ReLU and sigmoid networks exhibit behaviour that aligns more closely with experimentally observed psychophysical patterns. These observations suggest that biological implausibility of single-unit dynamics may translate into circuit mechanisms and behaviour that deviate from biological systems.

We observed that more biologically plausible ReLU and sigmoid RNNs show less heterogeneous single-unit selectivity under Dale's connectivity constraint compared with their unconstrained counterparts (Fig. 2c). Consistently, latent circuits of the same dimensionality captured more variance in Dale-constrained networks than in unconstrained ones (Table 1). Although Dale's constraint had relatively

modest effects, evaluating the impact of this fundamental feature of cortical circuits was essential. Dale's connectivity constraint introduces only a coarse level of biological realism, which may not offer a sufficient inductive bias to capture complex circuit solutions implemented in biological networks.

Although RNNs are coarse models of biological neural networks and their units' activation functions oversimplify single-neuron dynamics, our findings emphasize the critical role of the RNN architecture in generating biologically relevant hypotheses. RNN architectures carry inductive biases for the emergent task solutions, which can lead to distinct hypotheses about how the brain solves these tasks. While it remains an open question which RNN architectures best align with biological data, our results suggest that tanh activation function may not be the optimal choice. Directly comparing neural recording data with the representations and dynamics of RNNs across different

architectures will help determine which architecture best suits the modelling of biological neural networks.

## Methods

### RNN architectures and training procedure

For each of the six architectures ({Dale, No Dale} ⊗ {ReLU, sigmoid, tanh}), we trained 100 fully connected RNNs, each with $N = 100$ units, to solve cognitive tasks. The RNN dynamics are described by the equation

$$\tau \dot{\mathbf{y}} = -\mathbf{y} + f(W_{\mathrm{rec}}\mathbf{y} + W_{\mathrm{inp}}\mathbf{u}), \tag{1}$$

where $f$ is the activation function (ReLU, sigmoid or tanh). The sigmoid function is defined as $\mathrm{sigmoid}(\mathbf{x}) = 1/(1 + e^{-7.5\mathbf{x}})$ with the slope 7.5. The ReLU function is defined as $\mathrm{ReLU}(\mathbf{x}) = \max(0, \mathbf{x})$, and tanh function is $\tanh(\mathbf{x}) = (e^{\mathbf{x}} - e^{-\mathbf{x}})/(e^{\mathbf{x}} + e^{-\mathbf{x}})$.

The RNNs are trained by minimizing the loss function

$$\begin{aligned} \mathrm{Loss} \ = & \ \langle \| \mathbf{o}[\,\mathrm{mask}\,] - \hat{\mathbf{o}}[\,\mathrm{mask}\,] \|_2^2 \rangle + \lambda_r \langle \|\mathbf{y}\|_2^2 \rangle \\ & + \lambda_\perp \langle \| W_{\mathrm{inp}}^T W_{\mathrm{inp}} - \mathrm{diag}(W_{\mathrm{inp}}^T W_{\mathrm{inp}}) \|_2^2 \rangle. \end{aligned} \tag{2}$$

We initialize the RNN connectivity matrices as described previously[3]. In networks without Dale's constraint, the elements of the recurrent connectivity matrix were sampled from a Gaussian distribution $W_{ij}' \sim N(\mu, \sigma^2)$ with $\mu = 1/\sqrt{N}$, $\sigma = 1/N$. The spectral radius of the recurrent connectivity was then adjusted using the formula $W_{\mathrm{rec}} = \frac{\mathrm{s.r.}}{\max_k |\lambda_k|} W_{\mathrm{rec}}'$, where the new spectral radius s.r. = 1.2, and $\max_k |\lambda_k|$ is the eigenvalue of $W_{\mathrm{rec}}$ with the largest norm.

For networks with Dale's constraint, the weights were sampled differently for the excitatory or inhibitory units. We sampled excitatory weights as the absolute values of random variables drawn from a normal distribution $N(\mu_E, \sigma_E^2)$ with $\mu_E = 1/\sqrt{N}$, $\sigma_E = 1/N$. Inhibitory weights were sampled as the negative absolute values of random variables from $N(\mu_I, \sigma_I^2)$, with $\mu_I = R_{E/I}/\sqrt{N}$, $\sigma_I = 1/N$, where $R_{E/I}$ is the ratio of the number of excitatory and inhibitory neurons. We used $R_{E/I} = 4$ for ReLU and sigmoid RNNs and $R_{E/I} = 1$ for Dale-constrained tanh RNNs. We adjusted the spectral radius of the recurrent connectivity matrix using the same procedure as for the networks without Dale's constraint.

In all networks, the input $W_{\mathrm{inp}}$ and output $W_{\mathrm{out}}$ connectivity matrices were initialized by sampling raw values from a Gaussian distribution $N(\mu, \sigma^2)$, $\mu = 1/\sqrt{N}$, $\sigma = 1/N$ and then taking the absolute value of the elements to enforce non-negativity. Regardless of whether Dale's constraint was applied, the elements of $W_{\mathrm{inp}}$ and $W_{\mathrm{out}}$ were constrained to remain non-negative throughout training.

All connectivity matrices ($W_{\mathrm{inp}}$, $W_{\mathrm{rec}}$ and $W_{\mathrm{out}}$) were trained simultaneously using Adam optimizer in PyTorch, with the default hyperparameters: learning rate $\alpha = 0.001$, $\beta_1 = 0.9$, $\beta_2 = 0.999$, $\epsilon = 10^{-8}$. While training the networks with Dale's constraint, if any element of these matrices switched signs, it was set to zero to ensure that none of the constraints were violated.

The RNN output was obtained by running the RNN's dynamics forward for a given batch of inputs. We discretize the RNN dynamics using the first-order Euler scheme with a time-step d$t = 1$ ms and add a noise term in the discretized equation to obtain

$$\mathbf{y}_{t+1} = (1 - \gamma)\mathbf{y}_t + \gamma f\left(W_{\mathrm{rec}}\mathbf{y}_t + W_{\mathrm{inp}}\left(\mathbf{u}_t + \sqrt{2\gamma\sigma_{\mathrm{inp}}^2}\boldsymbol{\zeta}_t\right) + \sqrt{2\gamma\sigma_{\mathrm{rec}}^2}\boldsymbol{\xi}_t\right). \tag{3}$$

Here $\gamma = \mathrm{d}t/\tau$, and $\xi_t$ and $\zeta_t$ are random vectors with elements sampled from the standard normal distribution $N(0, 1)$. The hyperparameters used for RNN training are provided in Extended Data Table 3. RNNs were trained on the CDDM and Go/NoGo tasks with $\lambda_r = 0.5$ for $n_{\mathrm{iter}} = 5{,}000$ iterations. RNNs were trained on the memory number task first with $\lambda_r = 0$ for $n_{\mathrm{iter}} = 6{,}000$ and then with $\lambda_r = 0.3$ for additional $n_{\mathrm{iter}} = 6{,}000$. The code for RNN training is available as trainRNNbrain package via GitHub at https://github.com/engellab/trainRNNbrain (ref. 46).

### CDDM task

The task structure is presented in Fig. 1a. Two mutually exclusive context channels signal either 'motion' or 'colour' context. For a given context, a constant input with an amplitude of 1 is supplied through the corresponding channel for the entire trial duration. Sensory stimuli with two modalities ('motion' and 'colour') are each supplied through two corresponding input channels, encoding momentary evidence for choosing either the right or left response. Within each sensory modality, the mean difference between inputs on two channels represents the stimulus coherence, with values ranging from −1 to +1. During training, we used a discrete set of 15 coherences for each sensory modality: $c = \{0, \pm0.01, \pm0.03, \pm0.06, \pm0.13, \pm0.25, \pm0.5, \pm1\}$. The coherence $c$ was translated into two sensory inputs as $[(1 + c)/2, (1 - c)/2]$. For 300 time steps on a trial, the (6, 300)-dimensional input-stream array was calculated based on the triplet (binary context, motion coherence and colour coherence), generating $N_{\mathrm{batch}} = 2 \times 15 \times 15 = 450$ distinct trial conditions.

On each trial, the target output was set to 0 for each time step $t < 100$ ms. During the decision period $t > 200$ ms, the target was set as follows: if the relevant coherence (for example, coherence of 'motion' stimuli on a 'motion' context trial) was positive, the target for 'right' output channel was set to 1 from 200 ms onwards. If the relevant coherence was negative, the target for 'left' output channel was set to 1 instead. If the relevant coherence was 0, both output targets were set to 0. The target was specified for only a subset of time steps, forming a training mask (0 − 100) and (200 − 300) ms: enforcing no decision output before stimulus onset (0 − 100) ms and allowing the network to integrate stimulus without penalty before decision commitment during (200 − 300) ms.

### Go/NoGo and memory number tasks

The structure of these tasks is presented in Fig. 1b,c. For both tasks, we used 11 uniformly spaced input values $\mathcal{I}$, ranging from 0 to 1, delivered through the first input channel. The 'Go Cue' input is delivered through the second channel and activated only at time $t_{\mathrm{GoCue}}$ at the end of the trial, signalling that the RNN is required to respond. Finally, a constant bias input with an amplitude of 1 is supplied via the third channel throughout the entire trial duration. In the Go/NoGo task, the input value $\mathcal{I}$ was provided for the entire trial duration of 60 ms. The target output is determined as 0 before and $\Theta(\mathcal{I} - 0.5)$ after the Go Cue onset, where $\Theta$ is the Heaviside step function (Fig. 1b). If the input value was exactly 0.5, the network was required to output 0.5 after the Go Cue. In the memory number task, the input value $\mathcal{I}$ was present only for 10 ms, with the randomized stimulus onset time $t_{\mathrm{stim}} \sim U(0, 20)$ ms (Fig. 1c). The target output value was set to 0 before the Go Cue and the input value $\mathcal{I}$ afterwards. The onset of the Go Cue was set to $t_{\mathrm{GoCue}} = 30$ ms for the Go/NoGo task and $t_{\mathrm{GoCue}} = 70$ ms for the memory number task.

### RNNs with shuffled connectivity

For each of the analysed RNNs, we produced another RNN with shuffled connectivity as a control. To shuffle the connectivity, we randomly permute each row $R_i$ in the input matrix $W_{\mathrm{inp}}$ ($i$th row contains all inputs to unit $i$). We also randomly permute non-diagonal elements of each column in the recurrent matrix $W_{\mathrm{rec}}$ ($i$th column contains all outputs of unit $i$). We keep the diagonal elements in $W_{\mathrm{rec}}$ unchanged to preserve self-excitation of each unit.

### Analysis of population trajectories

We analysed 50 RNNs with the best task performance from each architecture. We simulated each RNN (including the corresponding control RNNs) to acquire a tensor of neural responses $Z$ with dimensionality $(N, T, K)$, where $N$ is the number of units in the network, $T$ is the number of time steps in a trial, and $K$ is the number of trials. We reshape the

neural response tensor $Z$ to obtain a matrix $X$ with dimensionality ($N$, $TK$). We then obtain a denoised matrix $F$ with dimensionality ($n_{PC}$, $TK$) by projecting matrix $X$ onto the first $n_{PC} = 10$ PCs along the first dimension, capturing more than 93% of variance in each instance across all RNNs and tasks. Reshaping matrix $F$ back into a three-dimensional tensor, we obtain a denoised tensor $\hat{Z}$ with dimensionality ($n_{PC}$, $T$, $K$) containing reduced population trajectories. We further normalized the reduced trajectory tensor $\hat{Z}$ by its variance, so that the reduced trajectory tensors have the same scale across all RNNs.

To obtain an MDS embedding of the reduced trajectories, we compute a distance matrix between reduced trajectory tensors $\hat{Z}_i$ and $\hat{Z}_j$ for each pair of RNNs $i$ and $j$. First, we obtain the optimal linear transformation between the matrices $F_i$ and $F_j$ corresponding to $\hat{Z}_i$ and $\hat{Z}_j$ using linear least squares regression with the function numpy.linalg.lstsq in python. We perform two regression analyses: first regressing $F_i$ onto $F_j$ and then $F_j$ onto $F_i$, resulting in two linear transformations $M_{ij}$ and $M_{ji}$, and two scores, score$_1 = \|F_i M_{ij} - F_j\|_2$ and score$_2 = \|F_j M_{ji} - F_i\|_2$. We then compute the distance between two trajectory tensors as the average of two scores: $d_{ij} = d_{ji} = (\text{score}_1 + \text{score}_2)/2$. We use these pairwise distances to compute MDS embedding with the function sklearn.manifold.MDS from sklearn package in python.

### Analysis of single-unit selectivity
For each RNN (including the control RNNs), we start with the same neural response tensor $Z$ as for the analysis of population trajectories. We reshape $Z$ to obtain matrix $X$ with dimensionality ($N$, $TK$). We then obtain a denoised matrix $G$ with dimensionality ($N$, $n_{PC}$) by projecting matrix $X$ onto the first $n_{PC} = 10$ PCs along the second dimension, capturing more than 90% of variance in each instance across all RNNs and tasks. We further normalize the resulting single-unit selectivity matrix $G$ by its variance, so that single-unit selectivity matrices have the same scale across all RNNs.

To obtain an MDS embedding, we compute a distance matrix between the single-unit selectivity matrices $G_i$ and $G_j$ for each pair of RNNs $i$ and $j$. To compute the distance between $G_i$ and $G_j$, we view each RNN unit as a point in $n_{PC}$-dimensional selectivity space. We then register the point configurations of two RNNs with an optimal orthogonal transformation that permits one-to-many mapping. To register the points, we use ICP registration algorithm ('ICP registration' section). Since there is no one-to-one correspondence between units in two RNNs, we perform the ICP registration two times: registering $G_i$ to $G_j$ and then $G_j$ to $G_i$, producing score$_1$ and score$_2$. We then set the distances $d_{ij} = d_{ji} = (\text{score}_1 + \text{score}_2)/2$. Since the ICP registration often converges to local minima, to register each pair of point clouds we run the registration procedure 60 times to ensure higher probability of accurate estimate of the distance between the two point clouds. We take the best result, corresponding to the minimal point cloud mismatch.

### Fixed-point finder
To find fixed points of an RNN, we use a custom fixed-point finder algorithm. For each constant input $\mathbf{u}$, we search for fixed points by minimizing the right-hand side in equation (1), $F(\mathbf{y}, \mathbf{u}) = -\mathbf{y} + f(W_{rec}\mathbf{y} + W_{inp}\mathbf{u})$ with scipy.optimize.fsolve function from scipy.optimize package in python. We accept point $\mathbf{y}^*$ as a fixed point if $\| F(\mathbf{y}^*, \mathbf{u})\|_2^2 \leqslant 10^{-12}$. The fsolve function also takes the Jacobian matrix $J(\mathbf{y}, \mathbf{u}) = \partial F(\mathbf{y}, \mathbf{u})/\partial \mathbf{y}$ of the RNN as an additional argument to enhance the efficiency of the optimization process. We initialize the minimization at a value $\mathbf{y}_0$ sampled randomly from the RNN trajectories: we choose a random trajectory $k$ from $K$ trials, and then a random time-step $t$ from the interval ($n_t/2, n_t$), that is, from the second half of the trial. We then add Gaussian noise $\xi \sim N(0, 0.01)$ to each coordinate of the sampled point to obtain the initial condition $\mathbf{y}_0$.

To find multiple fixed points for the same input $\mathbf{u}$, we search for fixed points starting from multiple initial conditions within an iterative loop. On each iteration of this loop, we sample a new initial condition

and perform the minimization to find a fixed point. We then compare this newly found fixed point $\mathbf{y}^*_{new}$ to all previously found fixed points $\mathbf{y}^*_{old}$. If the distance $\| \mathbf{y}^*_{new} - \mathbf{y}^*_{old}\|_2 \leqslant 10^{-7}$, then we discard the new fixed point because it lies too close to one of the previously found fixed points. This iterative loop continues until either 100 distinct fixed points were found in total or no new fixed points were found for 100 consecutive iterations.

We determine the fixed-point type (stable or unstable) by computing the principal eigenvalue $\lambda_0$ of the Jacobian $J(\mathbf{y}, \mathbf{u})$ evaluated at the fixed point. We classify the fixed point as stable if $\mathbb{R}e(\lambda_0) \leqslant 0$ and otherwise as unstable.

### Analysis of fixed points
For each RNN (including the control RNNs), we computed fixed points for each combination of input stimuli using a custom fixed-point finder algorithm ('Fixed-point finder' section), obtaining a fixed-point configuration, which is a set of stable and unstable fixed points for different combinations of inputs. We collect the coordinates of all fixed points in a matrix $P$ with dimensions ($N_p$, $N$), where $N_p$ is the total number of fixed points (both stable and unstable) across all the inputs and $N$ is the number of units. We reduce the second dimension of the matrix $P$ by projecting the fixed points onto the first $n_{PC} = 7$ PCs. We further normalized the resulting matrix by its variance, so that these fixed-point configurations have the same scale across all RNNs, obtaining a matrix $\hat{P}$ for each RNN. Throughout the transformations, we keep each fixed point tagged by its type and the corresponding input for which it was computed.

To obtain an MDS embedding, we compute a distance matrix between fixed-point configurations $\hat{P}_i$ and $\hat{P}_j$ for each pair of RNNs $i$ and $j$. To compute the distances between the two projected fixed-point configurations $\hat{P}_i$ and $\hat{P}_j$, we compute an optimal orthogonal transformation between the two sets of projected fixed points using orthogonal Procrustes with ICP registration ('ICP registration' section). When matching the projected fixed points, we restricted matches to the fixed points with the same tag (of the same type and obtained for the same input). We perform the ICP registration two times, registering $\hat{P}_i$ to $\hat{P}_j$ and then $\hat{P}_j$ to $\hat{P}_i$, resulting in two scores score$_1$ and score$_2$. We then set the distances $d_{ij} = d_{ji} = (\text{score}_1 + \text{score}_2)/2$. Using the distance matrix, we then obtain MDS embedding. To register each pair of point clouds, we run the registration procedure $n_{tries} = 60$ times and then take the result corresponding to the minimal fixed-point cloud mismatch.

### Analysis of trajectory endpoint configurations
For each RNN (including the control RNNs), we use the same neural response tensor $Z$ as for the analysis of population trajectories. We then restrict the data to the last time step of each trial, resulting in ($K$, $N$) dimensional matrix $S$ for each RNN containing the trajectory endpoint configuration. We further project the trial endpoints in $S$ onto first $n_{PC} = 10$ PCs, obtaining ($K$, $n_{PC}$)-dimensional matrix $\hat{S}$. Finally, we normalize each trajectory endpoint configuration matrix $\hat{S}$ by its variance, so that these endpoint configurations have the same scale across all RNNs. We compute the distance between two matrices $\hat{S}_i$ and $\hat{S}_j$ for RNNs $i$ and $j$ using the same procedure as for the population trajectory matrices $F$ ('Analysis of population trajectories' section). Using the distance matrix, we then obtain MDS embedding.

### ICP registration
To register the point clouds ('Analysis of single-unit selectivity' and 'Analysis of fixed points' sections), we use an ICP algorithm, which proceeds in four steps:

1. Initialization: define a random orthogonal matrix $A$ that transforms each point of the source point cloud $P_{source}$ into $P_{source} A$.
2. Point matching: For each point in the target point cloud $P_{target}$, find the closest point in the transformed source point cloud $P_{source} A$. Construct a new matrix $\hat{P}_{source}$ where the $i$th point is the

point from $P_{source} A$ closest to the $i$th point in $P_{target}$ (points in $\hat{P}_{source}$ may repeat).

3. Transformation update: update the transformation matrix $A$ to minimize the distance between $\hat{P}_{source}$ and $P_{target}$ using the orthogonal Procrustes method.

4. Iteration: repeat steps 2 and 3 until convergence.

This algorithm iteratively refines the transformation to achieve optimal alignment between the source and target point clouds. Since this optimization is non-convex, it may converge to a local optimum. Therefore, we perform each optimization for $n_{tries} = 60$ starting with random initializations and keep the solution with the minimal mean squared error as the distance between the source and target point clouds.

To compute distances between the fixed-point configurations ('Analysis of fixed points' section), we modify the point matching step by restricting possible matches only to the points obtained for the same inputs and of the same type (stable or unstable).

The code for the RNN analyses and the relevant datasets are available via GitHub at https://github.com/engellab/ActivationMattersRNN (ref. 47).

## Latent circuit inference

To identify the circuit mechanism supporting the CDDM task execution in an RNN, we fit its responses and task behaviour with the latent circuit model[12]. We model RNN responses $y$ as a linear embedding of dynamics $x$ generated by a low-dimensional RNN

$$\tau\dot{\mathbf{x}} = -\mathbf{x} + f(w_{rec}\mathbf{x} + w_{inp}\mathbf{u}), \qquad (4)$$

which we refer to as the latent circuit. Here $f$ is the activation function matching the activation function of the RNN. We also require the latent circuit to reproduce task behaviour via the output connectivity $w_{out}x$.

To fit the latent circuit model, we first sample RNN trajectories $Z$, forming a $(N, T, K)$-dimensional tensor. We then reduce the dimensionality of $Z$ using PCA to $N_{PC} = 30$, resulting in a tensor $z$ with $(N_{PC}, T, K)$ dimensions, capturing more than 99% of variance in $Z$ for all RNNs we analysed. We then infer the latent circuit parameters $w_{rec}, w_{inp}, w_{out}$ and an orthonormal embedding matrix $Q$ by minimizing the loss function

$$\text{Loss} = \langle \| \mathbf{o} - \hat{\mathbf{o}} \|_2^2 \rangle + \lambda_{emb}\langle \| Q\mathbf{x} - \mathbf{z} \|_2^2 \rangle + \lambda_w\left(\langle|w_{inp}|^2\rangle + \langle|w_{rec}|^2\rangle + \langle|w_{out}|^2\rangle\right) \qquad (5)$$

Here, $\langle \cdot \rangle$ denotes the mean over all dimensions of a tensor. Tensor $\mathbf{x}$ has the dimensionality $(n, T, K)$, where $n$ is the number of nodes in the latent circuit. This tensor $\mathbf{x}$ contains the activity of the latent circuit across $K$ trials and $T$ time steps per trial, and $\mathbf{y}$ is the corresponding activity tensor for the RNN. The $(N_{PC}, n)$ dimensional orthonormal matrix $Q$ embeds trajectories of the latent circuit $\mathbf{x}$ to match the RNN activity $\mathbf{z}$, such that $z \approx Qx$. Finally, $\mathbf{o}$ is the target circuit output, and the $\hat{\mathbf{o}} = w_{out}\mathbf{x}$ is the output produced by the latent circuit.

During optimization, we constrain the input matrix such that each input channel is connected to at most one latent node. To this end, we apply to the input matrix a mask, in which 1 indicates that the weight is allowed to change during training, and 0 indicates that the weight is fixed at 0. We design the mask such that each column has a single 1. Moreover, we constrain the elements of $w_{inp}$ and $w_{out}$ matrices to be non-negative.

We fitted latent circuit models to the ten RNNs with the best CDDM task performance from each architecture. For each RNN, we fit 8-node latent circuit model ≥30 times starting with random initializations and take the best-fitting circuit as a converged solution. The hyperparameters for the latent circuit fitting are provided in Extended Data Table 4. The code for latent circuit fitting is available via GitHub at https://github.com/engellab/latent_circuit_inference (ref. 48).

## Alignment of RNN dynamics with the output subspace

The norm of the readout matrix can affect the dynamics that emerge in RNNs through training[44]. In RNNs initialized with large readout norms, the network dynamics evolved in a subspace distinct from the output subspace spanned by the rows of the readout matrix[44]. The angle between the dynamics and output subspaces was large, and such dynamics were termed oblique. By contrast, in RNNs initialized with small readout norms, the angle between dynamics and readout subspaces was relatively small, and such dynamics were termed aligned.

In our networks, the weights of the output matrix were initialized with $\sigma = 1/N$, corresponding to the small readout norm, associated with aligned dynamics[44]. To quantify whether the resulting dynamics in our networks were aligned or oblique, we computed a generalized correlation measure $\rho$ (ref. 44), for the 50 top-performing networks of each architecture, during the epochs when RNNs were required to produce output. The generalized correlation measure $\rho = \frac{\|W_{out}^T X\|_F}{\|W_{out}\|_F \|X\|_F}$, where $X$ is the $(N, T_{out}, K)$ tensor with population activity of $N$ units during the task-epochs at which the networks were required to produce outputs ($T_{out}$ time steps in total) in $K$ trials; $\| \cdot \|_F$ refers to Frobenius norm.

We found that the dynamics in our networks lie along a continuum: neither fully aligned with the readout subspace nor strongly oblique (Extended Data Table 2). In addition, the generalized correlation measure $\rho$ was both task and architecture dependent. The dynamics were most aligned with the output subspace for ReLU networks trained on the Go/NoGo task. Furthermore, tanh networks tended to produce more oblique dynamics than sigmoid and ReLU RNNs. Since the initialization procedure and noise magnitude for inputs and recurrence were the same for all networks, this result further supports the conclusion that tanh networks rely on dynamics distinct from those of ReLU and sigmoid RNNs.

## Reporting summary

Further information on research design is available in the Nature Portfolio Reporting Summary linked to this article.

## Data availability

The synthetic data used in this study can be reproduced using the source code. The datasets containing the parameters of trained RNNs and the inferred latent circuits are available via GitHub at https://github.com/engellab/ActivationMattersRNN (ref. 47).

## Code availability

The code for RNN training is available as a trainRNNbrain package via GitHub at https://github.com/engellab/trainRNNbrain (ref. 46). The code for the analysis of the RNNs and the associated datasets are available via GitHub at https://github.com/engellab/ActivationMattersRNN (ref. 47). The code for latent circuit fitting is available via GitHub at https://github.com/engellab/latent_circuit_inference (ref. 48).

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

## Acknowledgements

This work was supported by National Institutes of Health (NIH) (grant no. RF1DA055666 to T.A.E.).

## Author contributions

P.T. and T.A.E. designed the research and developed the computational analysis framework. P.T. developed the code, performed computer simulations and analysed the data. P.T. and T.A.E wrote the paper.

## Competing interests

The authors declare no competing interests.

## Additional information

**Extended data** is available for this paper at https://doi.org/10.1038/s42256-025-01127-2.

**Correspondence and requests for materials** should be addressed to Tatiana A. Engel.

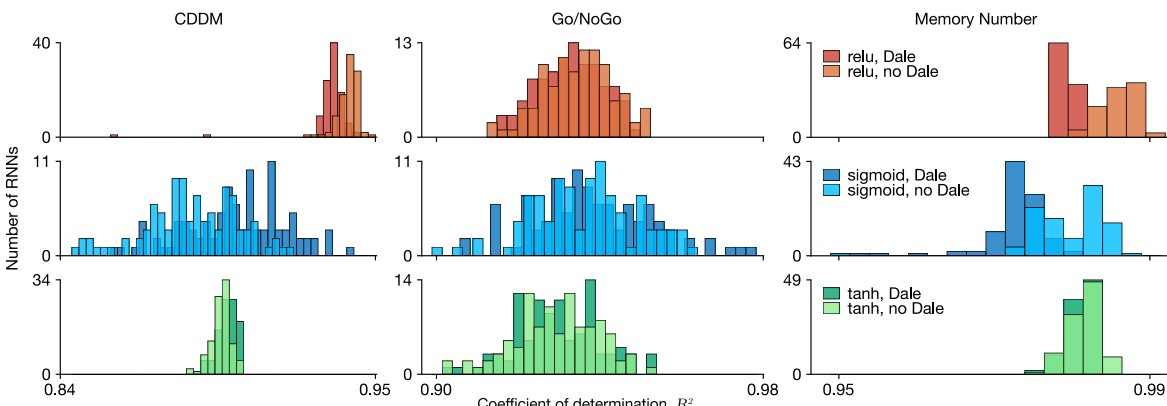

**Extended Data Fig. 1 | Distribution of task performance across RNNs.**
Histograms of the performance metric across 100 RNNs for each architecture, trained on CDDM, Go/NoGo and Memory Number tasks (bin size is 0.0025 in all plots). The performance metric is the coefficient of determination $R^2$ between RNN output and targets on test data. The histograms indicate that RNNs with different architectures achieve comparable task performance.

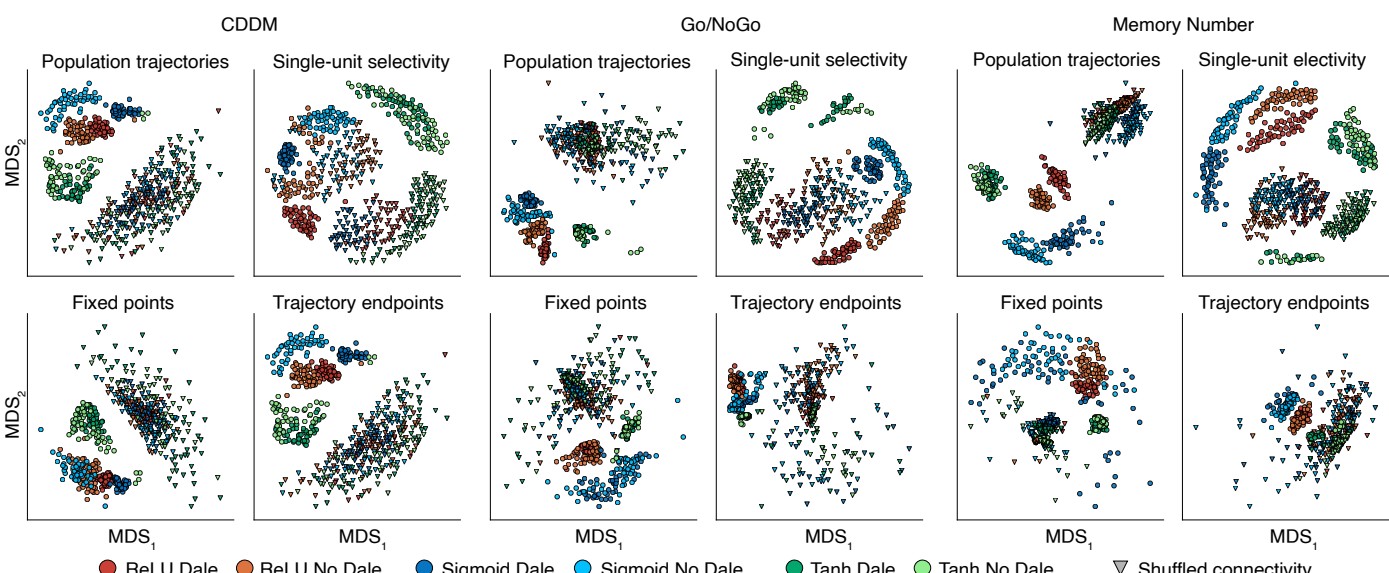

**Extended Data Fig. 2 | MDS embeddings of 50 lowest-performing RNNs across all tasks and architectures.** MDS embeddings of population trajectories, single-unit selectivity, fixed points and trajectory endpoint configurations across the tasks for the 50 lowest-performing RNNs, alongside the same networks with shuffled connectivity. The MDS embedding results are qualitatively similar between the 50 lowest-performing and 50 top-performing networks (cf. Fig. 2b,d, Fig. 3b,d, Fig. 5c,d).

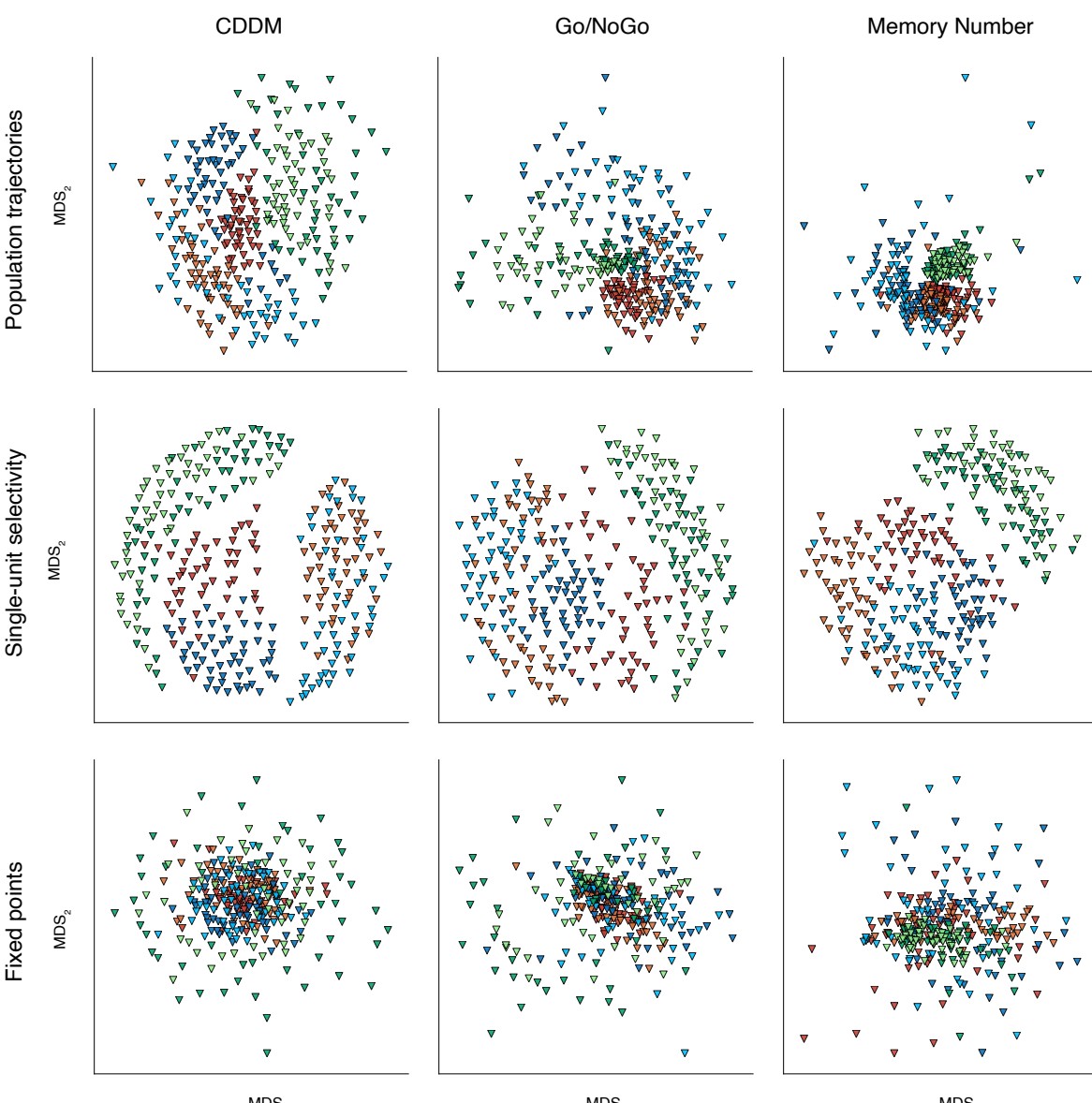

**Extended Data Fig. 3 | MDS embeddings of RNNs with shuffled connectivity.**
MDS embeddings of population trajectories, single-unit selectivity, and
fixed point configurations from the 50 top-performing RNNs with shuffled
connectivity. RNNs with shuffled connectivity and different activation functions
form distinct clusters in the embedding space of their population trajectories
and single-unit selectivity configurations. In contrast, the MDS embedding of
their fixed point configurations shows no clear separation, with all architectures
appearing largely indistinguishable.

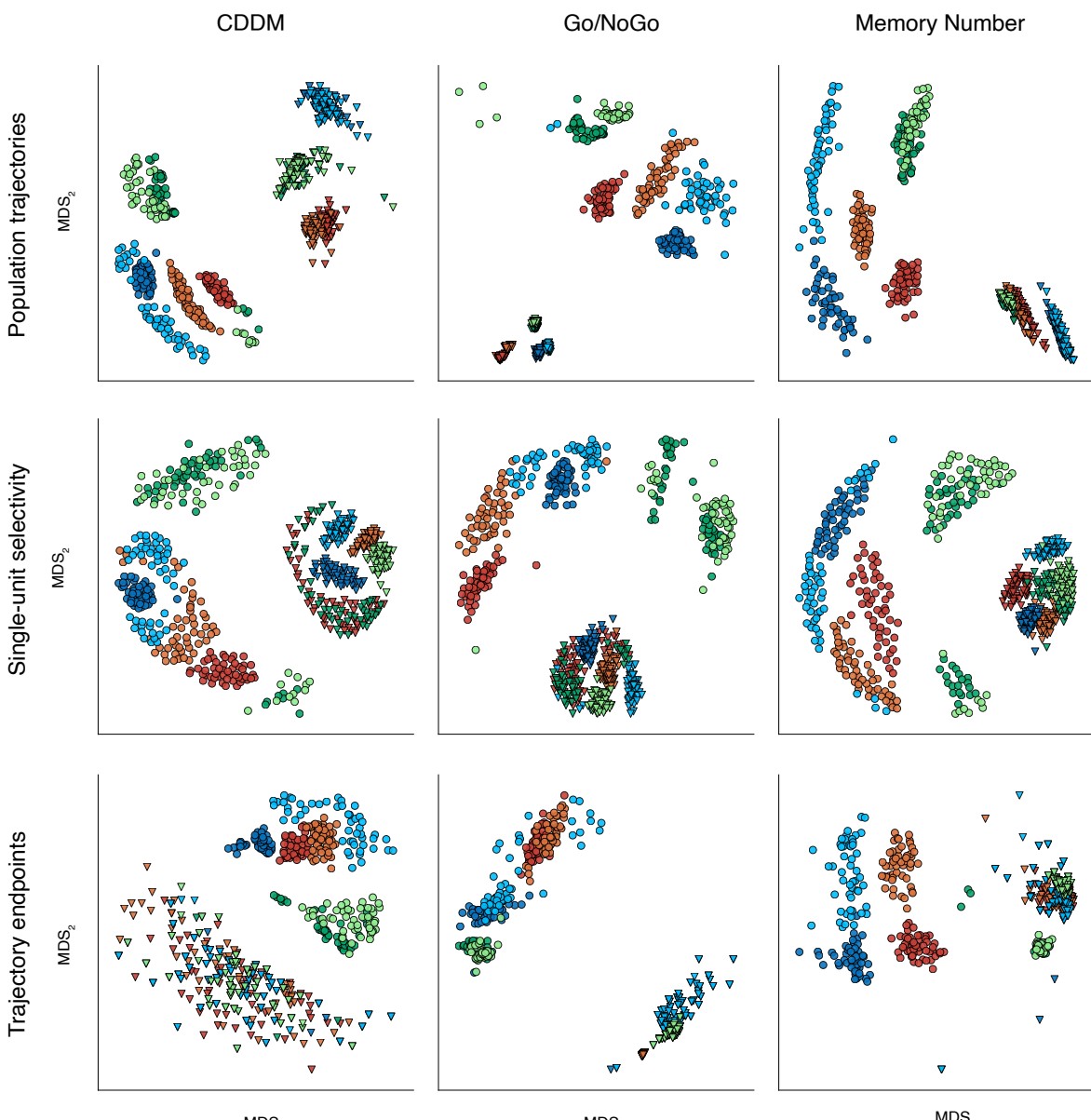

**Extended Data Fig. 4 | MDS embeddings of randomly initialized, untrained RNNs.** MDS embeddings of population trajectories, single-unit selectivity and trajectory endpoint configurations across tasks for 50 top-performing networks, alongside 50 randomly initialized and untrained networks from each architecture. The MDS embeddings of trained RNNs alongside randomly initialized untrained networks closely resemble the results seen in RNNs with shuffled connectivity (cf. Fig. 2b,d, Fig. 3b,d, Fig. 5c,d, Extended Data Fig. 3). The differences between the randomly initialized networks are evident in the embeddings of trajectories and selectivity configurations. In contrast, trajectory endpoint configurations (and, by extension, fixed point configurations) of untrained networks with different architectures remain indistinguishable, as they primarily reflect the shared input structure across networks.

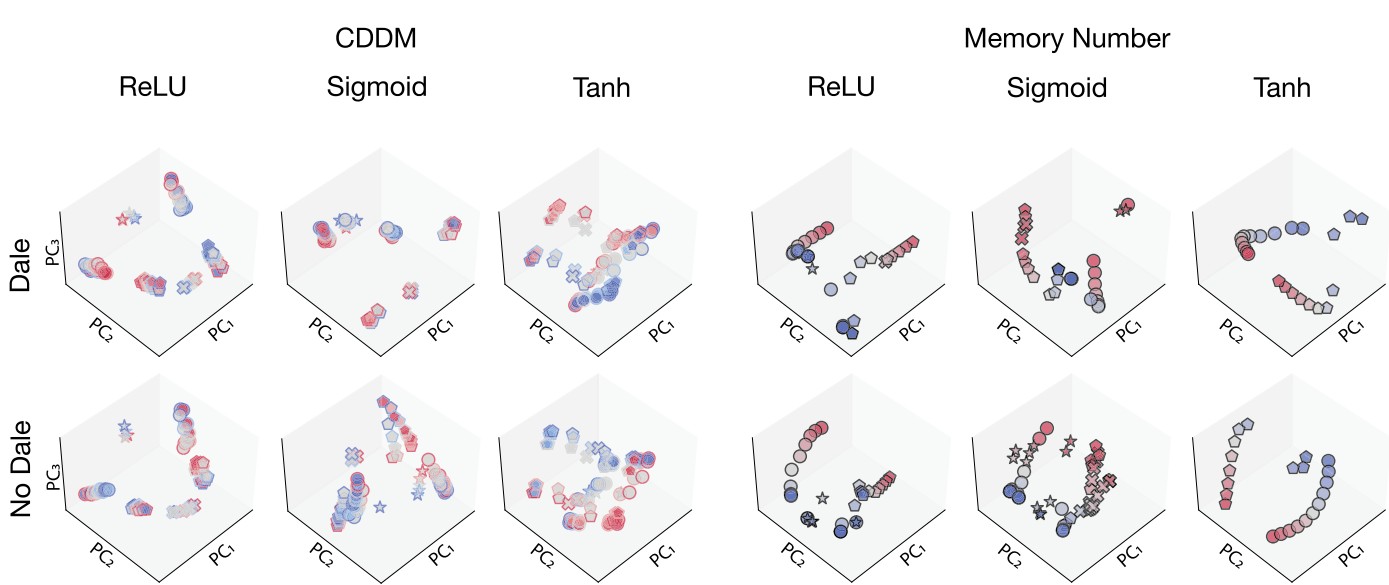

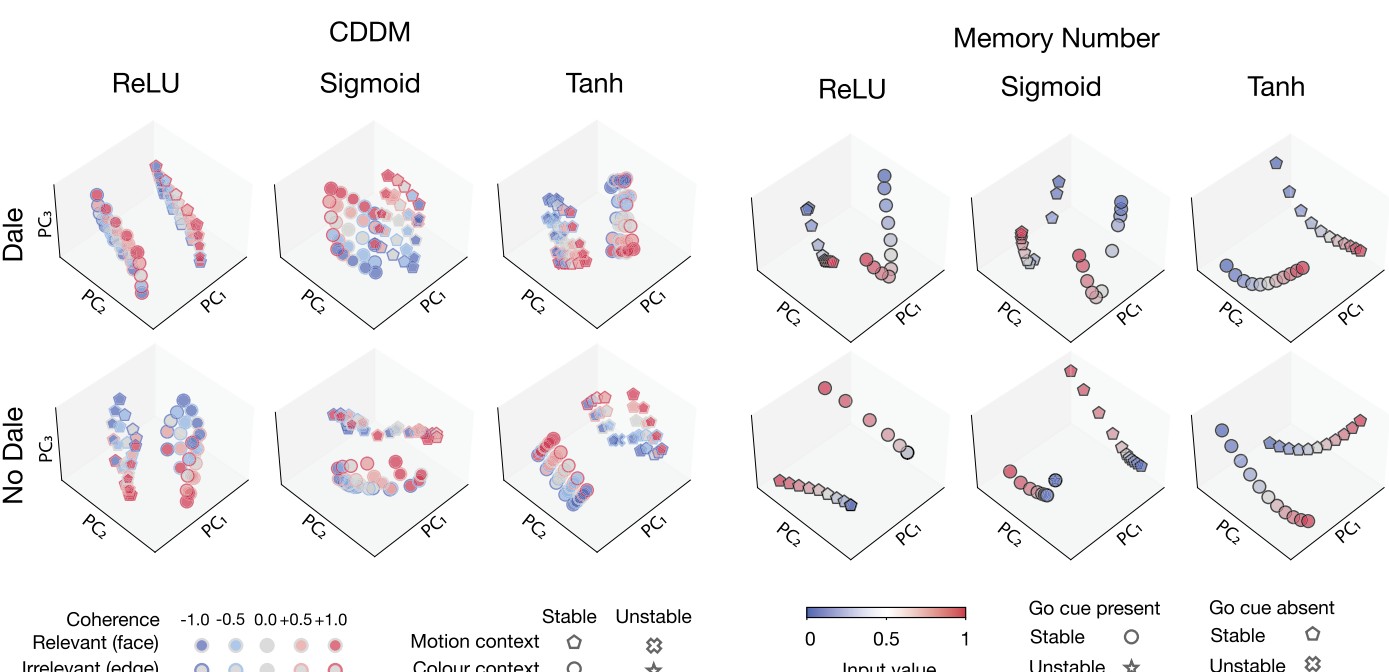

**Extended Data Fig. 5 | Fixed point configurations in trained and untrained networks.** Fixed point configurations in example trained and untrained networks across architectures for the Context-Dependent Decision-Making and Memory Number tasks. Trained networks exhibit distinct fixed point configurations depending on their activation function: ReLU and sigmoid RNNs display fixed-point arrangements that differ from those of tanh networks. In contrast, untrained networks exhibit nearly identical fixed point configurations across activation functions. This similarity arises because untrained networks largely preserve the geometry of task inputs in the activity of recurrent units, introducing minimal distortion.

**Extended Data Table 1 | Coefficient of determination $R^2$ between the RNN output and targets on test data**

|  | ReLU | | Sigmoid | | Tanh | |
|---|---|---|---|---|---|---|
|  | Dale | No Dale | Dale | No Dale | Dale | No Dale |
| CDDM | $94 \pm 0.2\%$ | $94 \pm 0.1\%$ | $92 \pm 0.9\%$ | $90 \pm 0.8\%$ | $90 \pm 0.1\%$ | $90 \pm 0.2\%$ |
| Go/NoGo | $94 \pm 0.4\%$ | $94 \pm 0.4\%$ | $95 \pm 1.0\%$ | $95 \pm 0.7\%$ | $94 \pm 0.6\%$ | $94 \pm 0.5\%$ |
| Memory Number | $98 \pm 0.1\%$ | $99 \pm 0.1\%$ | $98 \pm 0.1\%$ | $98 \pm 0.1\%$ | $98 \pm 0.1\%$ | $98 \pm 0.1\%$ |

Data are mean ± std across 50 top-performing RNNs for each task and architecture.

**Extended Data Table 2 | Alignment of RNN dynamics with the output subspace**

|  | ReLU | | Sigmoid | | Tanh | |
|---|---|---|---|---|---|---|
|  | Dale | No Dale | Dale | No Dale | Dale | No Dale |
| CDDM | $0.48 \pm 0.01$ | $0.48 \pm 0.02$ | $0.42 \pm 0.02$ | $0.50 \pm 0.03$ | $0.30 \pm 0.02$ | $0.28 \pm 0.02$ |
| Go/NoGo | $0.72 \pm 0.01$ | $0.71 \pm 0.01$ | $0.68 \pm 0.02$ | $0.66 \pm 0.03$ | $0.39 \pm 0.04$ | $0.42 \pm 0.04$ |
| Memory Number | $0.56 \pm 0.01$ | $0.55 \pm 0.01$ | $0.58 \pm 0.02$ | $0.58 \pm 0.03$ | $0.45 \pm 0.01$ | $0.45 \pm 0.02$ |

The generalized correlation measure $\rho$ between the RNN activity $X$ and output subspace across tasks for 50 top-performing networks from each architecture. The generalized correlation measure[44] is defined as $\rho = \frac{\|W_{out}^T X\|_F}{\|W_{out}\|_F \|X\|_F}$, where $X$ is ($N$, $T_{out}$, $K$) -dimensional activity matrix, $N$ is the number of units in the RNN, $T_{out}$ is the number of time steps in the trial for which output was required, $K$ is the number of different trials, and $\| \cdot \|_F$ represents Frobenius norm. Each row of the activity matrix $X$ contains activity of a single unit with the mean both over trials and times steps subtracted, so that each row of $X$ has zero mean.

**Extended Data Table 3 | RNN training hyperparameters**

| Parameter | Values |
|---|---|
| number of units N | 100 |
| dt | 1 ms |
| $\tau$ | 10 ms |
| Spectral radius | 1.2 |
| Learning rate | 0.005 |
| Weight decay | $5 \cdot 10^{-6}$ |
| $\sigma_{\mathrm{rec}}$ | 0.05 |
| $\sigma_{\mathrm{inp}}$ | 0.05 |
| $\lambda_{\perp}$ | 0.3 |
| $\lambda_r$ | 0.5 |
| $n_{\mathrm{iter}}$ | CDDM, Go/NoGo: 5,000<br>Memory Number: 6,000 with $\lambda_r = 0$, then 6,000 with $\lambda_r = 0.3$ |
| $n_t$ | CDDM: 300 ms<br>Go/NoGo: 60 ms<br>Memory Number: 120 ms |
| Mask | CDDM: $(0 - 100)$ & $(200 - 300)$ ms<br>Go/NoGo: $(10 - 30)$ & $(40 - 60)$ ms<br>Memory Number: $(0 - 60)$ & $(80 - 120)$ ms |

.

**Extended Data Table 4 | Hyperparameters for fitting latent circuit model**

| Parameter | Values |
|---|---|
| $N_{\mathrm{PC}}$ | 30 |
| dt | 1ms |
| $\tau$ | 10 ms |
| $n_{\mathrm{iter}}$ | 2,000 |
| Learning rate | 0.02 |
| $\lambda_w$ | 0.08 |

# Reporting Summary

## Statistics

For all statistical analyses, confirm that the following items are present in the figure legend, table legend, main text, or Methods section.

| n/a | Confirmed | |
|---|---|---|
| ☐ | ☒ | The exact sample size (*n*) for each experimental group/condition, given as a discrete number and unit of measurement |
| ☐ | ☒ | A statement on whether measurements were taken from distinct samples or whether the same sample was measured repeatedly |
| ☒ | ☐ | The statistical test(s) used AND whether they are one- or two-sided<br>*Only common tests should be described solely by name; describe more complex techniques in the Methods section.* |
| ☒ | ☐ | A description of all covariates tested |
| ☒ | ☐ | A description of any assumptions or corrections, such as tests of normality and adjustment for multiple comparisons |
| ☐ | ☒ | A full description of the statistical parameters including central tendency (e.g. means) or other basic estimates (e.g. regression coefficient) AND variation (e.g. standard deviation) or associated estimates of uncertainty (e.g. confidence intervals) |
| ☒ | ☐ | For null hypothesis testing, the test statistic (e.g. *F*, *t*, *r*) with confidence intervals, effect sizes, degrees of freedom and *P* value noted<br>*Give P values as exact values whenever suitable.* |
| ☒ | ☐ | For Bayesian analysis, information on the choice of priors and Markov chain Monte Carlo settings |
| ☒ | ☐ | For hierarchical and complex designs, identification of the appropriate level for tests and full reporting of outcomes |
| ☒ | ☐ | Estimates of effect sizes (e.g. Cohen's *d*, Pearson's *r*), indicating how they were calculated |

*Our web collection on statistics for biologists contains articles on many of the points above.*

## Software and code

Policy information about availability of computer code

| | |
|---|---|
| Data collection | Synthetic data were generated using custom Python 3 code. The source code to reproduce results of this study is freely available on GitHub: https://github.com/engellab/trainRNNbrain , https://github.com/engellab/latent_circuit_inference , https://github.com/engellab/ActivationMattersRNN . The analyzed synthetic data are available at https://github.com/engellab/ActivationMattersRNN/tree/main/data |
| Data analysis | Data analysis was performed using custom Python 3 code available at https://github.com/engellab/ActivationMattersRNN. |

For manuscripts utilizing custom algorithms or software that are central to the research but not yet described in published literature, software must be made available to editors and reviewers. We strongly encourage code deposition in a community repository (e.g. GitHub). See the Nature Portfolio guidelines for submitting code & software for further information.

## Data

Policy information about availability of data

All manuscripts must include a data availability statement. This statement should provide the following information, where applicable:
- Accession codes, unique identifiers, or web links for publicly available datasets
- A description of any restrictions on data availability
- For clinical datasets or third party data, please ensure that the statement adheres to our policy

All synthetic data reported in this paper can be reproduced using the source code. The data are also available at https://github.com/engellab/ActivationMattersRNN/tree/main/data.

## Human research participants

Policy information about studies involving human research participants and Sex and Gender in Research.

| | |
|---|---|
| Reporting on sex and gender | NA |
| Population characteristics | NA |
| Recruitment | NA |
| Ethics oversight | NA |

Note that full information on the approval of the study protocol must also be provided in the manuscript.

# Field-specific reporting

Please select the one below that is the best fit for your research. If you are not sure, read the appropriate sections before making your selection.

☒ Life sciences          ☐ Behavioural & social sciences          ☐ Ecological, evolutionary & environmental sciences

For a reference copy of the document with all sections, see nature.com/documents/nr-reporting-summary-flat.pdf

# Life sciences study design

All studies must disclose on these points even when the disclosure is negative.

| | |
|---|---|
| Sample size | We trained 100 RNNs with each architecture ({Dale, No Dale} x {ReLU, tanh, sigmoid}) on each task, providing a robust sample size for our analyses. No statistical methods were used to predetermine sample size. The sample size used here substantially exceeds that of similar previous studies; for example, Maheswaranathan et al. (NeurIPS 2019) used 30 RNNs per architecture. |
| Data exclusions | No data were excluded |
| Replication | We trained 100 RNNs with each architecture on each task. The results were replicated between the top 50 RNNs with the best task performance and the remaining 50 RNNs. |
| Randomization | Randomization is not relevant to this study. RNNs were not allocated into groups. All RNNs were analyzed using the same processing functions and procedures. |
| Blinding | Blinding is not relevant to this study. RNNs were not allocated into groups. All RNNs were analyzed using the same processing functions and procedures. |

# Reporting for specific materials, systems and methods

We require information from authors about some types of materials, experimental systems and methods used in many studies. Here, indicate whether each material, system or method listed is relevant to your study. If you are not sure if a list item applies to your research, read the appropriate section before selecting a response.

### Materials & experimental systems

| n/a | Involved in the study |
|---|---|
| ☒ | ☐ Antibodies |
| ☒ | ☐ Eukaryotic cell lines |
| ☒ | ☐ Palaeontology and archaeology |
| ☒ | ☐ Animals and other organisms |
| ☒ | ☐ Clinical data |
| ☒ | ☐ Dual use research of concern |

### Methods

| n/a | Involved in the study |
|---|---|
| ☒ | ☐ ChIP-seq |
| ☒ | ☐ Flow cytometry |
| ☒ | ☐ MRI-based neuroimaging |

