## [Peer Review File · Nature Machine Intelligence]

Single-unit activations confer inductive biases for emergent circuit solutions to cognitive tasks

Corresponding Author: Professor Tatiana Engel

Version 0:

Reviewer comments:

Reviewer #1

(Remarks to the Author)

Tolmachev and Engel illustrate very clearly in this article that the solutions that RNNs find when trained on cognitive tasks are not unique, and may depend strongly on relatively understudied details, such as the particular input-output function used, or the imposed connectivity constraints such as Dale's law.

This issue is very important, especially for the field of neuroscience, raising a valid point that training an RNN on a task is not sufficient to understand the underlying neural mechanism in the brain. While the results shown in this work are not very surprising, especially for people with some experience training RNNs, it is a valid and important point that should be made.

One problem with this type of approach is that it is unclear what makes a solution different and what makes it similar. The authors here use multiple approaches: principal component analysis of trajectories and neural tuning, fixed point analysis, and low-dimensional reduction of dynamics. While each approach has clear flaws, and some of the results lean towards being very descriptive, the accumulation of all the approaches them make the results quite compelling. I wish the authors spent more time trying to figure out why the differences emerge, but I guess that will be part of future research.

(Relatively) major issues:

One thing that is very surprising is that tanh, sigmoid and relu networks seem indistinguishable when the connectivity is shuffled (Fig 3B and D, and also Fig 5). This seems quite strange, since generally these networks will have very different properties, even without any training. Why is this the case?

Related to the previous issue, the Methods are not very rigorous, and some key information is missing. In particular, how are the networks initialized? What are the initial connectivity matrices and the initial readout? It is known, for instance, that the norm of the readout will affect very strongly the solutions of RNNs (Schuessler et al. 2024, eLife). What is the norm? It would be interesting to perform the MDS before training the RNNs (maybe as a supplementary figure), to understand whether the differences observed in the solutions are only emerging through the training procedure.

Minor issues:

Why are the authors interested in Dale's law at all? I believe this choice deserves a more extended motivation. There are many hyperparameters and constraints that can be added to the RNN, and that will likely change solutions. Why this one in particular?

Methods: what is trained in the RNNs? Readout and input weights are trained?

Methods: alpha is used twice (learning rate and dt/tau)

(Remarks on code availability)

I reviewed the code just very briefly. I couldn't easily find where the code for training RNNs is (maybe in the yaml files?). I am not sure if it has not been shared, and only the RNN data is shared, or if I couldn't find it.

Reviewer #2

(Remarks to the Author)

This study is a carefully considered, important analysis of the biases that different activation functions endow RNN solutions to various tasks. Rather than just highlighting the differences, this study also tries to identify why different activation functions have such different solutions. The latent circuit inference and the evaluation of how the networks handle out-of-distribution generalization provided valuable insights into their underlying mechanisms. The findings from both analyses underscored the significance of this study, highlighting the critical role of activation functions in network design and the potential risks of overlooking them. The inclusion of networks with and without Dale's Law was a good addition to the paper and provided a good benchmark for comparison of the size of the effects as noted by the authors at the end of the results section. The discussion was excellent. Given the biological motivation of the paper, I particularly appreciated the extensive comparison to experimental data and consideration of what may be biologically plausible.

Minor points:

- The manuscript states that the top 30 RNNs from each network architecture were selected for analysis. Providing insight into the performance of the remaining 70 networks would strengthen the paper and clarify the significance of this cutoff in shaping the results.
- There is a brief mention of implications for RNNs fit to reproduce neural data. Whilst I understand it is outside of the scope of this work to directly examine the effect of different activation functions on these methods, a little more detail here would be beneficial to the discussion.
- Some more thought might be given in either the discussion or the results as to other activation functions and whether each activation function causes unique network behavior or rather there are different groups (such as the similarities highlighted between ReLU and sigmoid networks in this paper).

(Remarks on code availability)

I have not run the code but I have looked through the repository and it seems well documented with a decent README.

Version 1:

Reviewer comments:

Reviewer #1

(Remarks to the Author)

The authors have addressed in details all my concerns with the previous version of the manuscript. Mainly, the Methods are now complete and it has been clarified what part of the effects shown in the study can be attributed to learning, and which effects correspond to differences present in untrained networks.

I believe this is a very useful study for the community. It analyzes in detail how different structural properties can drastically change the emergent properties in recurrent networks.

(Remarks on code availability)

Reviewer #2

(Remarks to the Author)

Thank you for addressing some of my points in the discussion and extending the analyses to the top 50 networks and the MDS to all the networks.

Thank you for the detailed explanation clarifying that similarities between activation functions appear to be task-dependent, with predictable relationships arising from the interaction between activation properties and specific computational demands.

As a suggestion for strengthening the manuscript, it might be valuable to consider whether there are systematic ways to determine, from task properties themselves, whether a task requires strongly positive-only activations (e.g., benefits from positivity constraints) or relies on saturation effects. Identifying measurable features or computational demands of a task that predict when positivity or saturation properties become critical could help establish more general principles linking task characteristics to activation-function similarities—and clarify how one could know ahead of time which tasks are likely to depend on specific activation properties. I think this would be valuable for anyone inspired by this thought-provoking work to change their approach to this type of modelling.

(Remarks on code availability)

I have not run the code but I have looked through the repository and it seems well documented with a decent README.

Point-by-point responses.

Reviewer #1 (Remarks to the Author):

Tolmachev and Engel illustrate very clearly in this article that the solutions that RNNs find when trained on cognitive tasks are not unique, and may depend strongly on relatively understudied details, such as the particular input-output function used, or the imposed connectivity constraints such as Dale's law.

This issue is very important, especially for the field of neuroscience, raising a valid point that training an RNN on a task is not sufficient to understand the underlying neural mechanism in the brain. While the results shown in this work are not very surprising, especially for people with some experience training RNNs, it is a valid and important point that should be made.

One problem with this type of approach is that it is unclear what makes a solution different and what makes it similar. The authors here use multiple approaches: principal component analysis of trajectories and neural tuning, fixed point analysis, and low-dimensional reduction of dynamics. While each approach has clear flaws, and some of the results lean towards being very descriptive, the accumulation of all the approaches them make the results quite compelling. I wish the authors spent more time trying to figure out why the differences emerge, but I guess that will be part of future research.

Reply:

We thank the reviewer for this enthusiastic assessment and recognition of the broad significance of our results for the field of neuroscience.

(Relatively) major issues:

One thing that is very surprising is that tanh, sigmoid and relu networks seem indistinguishable when the connectivity is shuffled (Fig 3B and D, and also Fig 5). This seems quite strange, since generally these networks will have very different properties, even without any training. Why is this the case?

Reply:

We agree that networks with different activation functions are expected to exhibit distinct properties, even when they are unable to perform the task—whether because they are untrained or due to shuffled connectivity in trained networks. Accordingly, RNNs with shuffled connectivity but different activation functions separate from each other in the MDS embeddings of single-unit selectivity configurations, and to a lesser extent, in the embeddings of population trajectories (Fig. 2B,D, Fig. 5C,D). However, as the reviewer points out, these same networks remain largely indistinguishable in the MDS embeddings of fixed-point and trajectory endpoint configurations (Fig. 3B,D, Fig. 5C,D). Fixed-point configurations poorly distinguish these networks because, in models with shuffled connectivity, fixed points are primarily determined by the input structure shared across models, regardless of activation function (Extended Data Fig. 3). Similarly, the configurations of trajectory endpoints, which closely mirror the fixed points, are shaped by inputs and do not depend on the choice of activation function.

To more clearly reveal differences among shuffled RNNs with varying architectures, we performed MDS embedding of population trajectories and single-unit selectivity configurations exclusively on RNNs with shuffled connectivity, excluding trained networks (Extended Data Fig. 4). Analyzed separately, these networks exhibit clear separation in both population trajectories and selectivity embeddings.

To summarize, distinctions between different architectures are evident in both randomly initialized networks and trained networks with shuffled connectivity, particularly when comparing population trajectories and single-unit selectivity. These differences become further amplified through training. However, RNNs with different architectures appear indistinguishable when analyzing fixed points and trajectory endpoint configurations, as these are primarily shaped by the geometry of inputs.

We present these analyses in new Extended Data Figs. 3,4 and on lines 97–99 and lines 134–137 in revised Results.

Related to the previous issue, the Methods are not very rigorous, and some key information is missing. In particular, how are the networks initialized? What are the initial connectivity matrices and the initial readout?

Reply:

We initialized the RNN connectivity matrices as in Song et. al, *PLoS Comp Biol* (2016). In networks without Dale's constraint, the elements of the recurrent connectivity matrix were sampled from a Gaussian distribution

$W_{rec\ ij} \sim N(\mu, \sigma^2)$ with $\mu = \frac{1}{\sqrt{N}}$, $\sigma = \frac{1}{N}$, where N is the number of units. The spectral radius of the recurrent

connectivity was then adjusted using formula $W_{rec} = \frac{sr}{\max_k |\lambda_k|} W'_{rec}$, where new spectral radius $sr = 1.2$, and

$\max_k |\lambda_k|$ is the eigenvalue of W'_{rec} with the largest norm.

For networks with Dale's constraint, the weights were sampled differently for the excitatory or inhibitory units. We sampled excitatory weights as the absolute values of random variables drawn from a normal distribution $N(\mu_E, \sigma_E^2)$

with $\mu_E = \frac{1}{\sqrt{N}}$, $\sigma_E = \frac{1}{N}$. Inhibitory weights were sampled as the negative absolute values of random variables from

$N(\mu_I, \sigma_I^2)$, with $\mu_I = \frac{R_{E/I}}{\sqrt{N}}$, $\sigma_I = \frac{1}{N}$, where, $R_{E/I}$ is the ratio between the number of excitatory and inhibitory neurons.

We used $R_{E/I} = 4$ for ReLU and sigmoid RNNs, and $R_{E/I} = 1$ for Dale-constrained tanh RNNs. We adjusted the spectral radius of the recurrent connectivity matrix using the same procedure as for the networks without Dale's constraint.

In all networks, the input W_{inp} and output W_{out} connectivity matrices were initialized by sampling raw values from a Gaussian distribution $N(\mu, \sigma^2)$, $\mu = \frac{1}{\sqrt{N}}$, $\sigma = \frac{1}{N}$, and then taking the absolute value of the elements to enforce non-negativity. Regardless of whether Dale's constraint was applied, the elements of the W_{inp} and W_{out} were constrained to remain non-negative throughout training.

All connectivity matrices (W_{inp} , W_{rec} , and W_{out}) were trained simultaneously using Adam optimizer in Pytorch, with the default hyperparameters: learning rate $\alpha = 0.001$, $\beta_1 = 0.9$, $\beta_2 = 0.999$, $\epsilon = 10^{-8}$, and weight decay parameter (corresponding to l_2 regularization of weights) set to 5×10^{-6} . While training the networks with Dale's constraint, if any element of these matrices switched signs, it was set to zero to ensure that none of the constraints were violated.

We added this information on lines 311–330 in revised Methods.

It is known, for instance, that the norm of the readout will affect very strongly the solutions of RNNs (Schuessler et al. 2024, eLife). What is the norm?

Reply:

We agree that the norm of the readout matrix makes a difference. Schuessler et al. (*eLife*, 2024) found that in RNNs initialized with large readout norms, network dynamics evolved in a subspace distinct from the output subspace spanned by the rows of the readout matrix. The angle between the dynamics and output subspaces was large, and such dynamics were termed *oblique*. In contrast, in RNNs initialized with small readout norms, the angle between dynamics and readout subspaces was relatively small, and such dynamics were termed *aligned*.

In our networks, the weights of the output matrix were initialized with $\sigma = \frac{1}{N}$, corresponding to the small readout norm in Schuessler et al. (*eLife*, 2024), associated with aligned dynamics. Similar to Schuessler et al., we trained all connectivity matrices (W_{inp} , W_{rec} , and W_{out}), applying an l_2 -norm penalty to discourage large weights. To quantify whether the resulting dynamics in our networks were aligned or oblique, we computed a generalized correlation measure ρ from Schuessler et al. for the 50 top-performing networks from each architecture, during the epochs when RNNs were required to produce output.

We found that the dynamics in our networks lie along a continuum—neither fully aligned with the readout subspace nor strongly oblique (Extended Data Table 4). In addition, the generalized correlation measure ρ was both task and

architecture dependent. The dynamics were most aligned with the output subspace for ReLU networks trained on the Go/NoGo task. Furthermore, tanh networks tended to produce more oblique dynamics than sigmoid and ReLU RNNs. Since the initialization procedure and noise magnitude for inputs and recurrence were the same for all networks, this result further supports the conclusion that tanh networks rely on dynamics distinct from those of ReLU and sigmoid RNNs.

We present these analyses in new Extended Data Table 4, on lines 268–271 in revised Discussion, and on lines 508–527 in revised Methods.

It would be interested to perform the MDS before training the RNNs (maybe as a supplementary figure), to understand whether the differences observed in the solutions are only emerging through the training procedure.

Reply:

We performed MDS embeddings of population trajectories, single-unit selectivity and trajectory endpoint configurations for both trained networks and randomly initialized RNNs to test whether the observed differences arise only through training (Extended Data Fig. 5). We did not include fixed-point configurations, as they closely resemble trajectory endpoint configurations but are much more computationally demanding to compute.

Randomly initialized untrained networks showed similar results to RNNs with shuffled connectivity (cf. Extended Data Fig. 4). Differences between RNNs with different activation functions are evident in trajectories and selectivity configurations even in untrained networks. The training further amplifies these distinctions. In contrast, trajectory endpoint configurations—and, by extension, fixed-point configurations—of untrained networks with different architectures remain indistinguishable, as they primarily reflect the input structure, similar to RNNs with shuffled connectivity.

We present these analyses in new Extended Data Fig. 5 and on lines 97–99 and lines 134–137 in revised Results.

Minor issues:

Why are the authors interested in Dale's law at all? I believe this choice deserves a more extended motivation. There are many hyperparameters and constraints that can be added to the RNN, and that will likely change solutions. Why this one in particular?

Reply:

Dale's principle is a fundamental feature of cortical networks, and incorporating it into models improves their biological plausibility. We therefore investigated whether Dale's constraint influences task solutions that emerge in recurrent neural networks through training. Although the effects were relatively minor, it was important to evaluate how this key biological feature shapes network dynamics and representations.

We clarified this point on line 27 in the revised Introduction and on lines 280–286 in revised Discussion.

Methods: what is trained in the RNNs? Readout and input weights are trained?

Reply:

Yes, the input, recurrent, and output connectivity matrices are trained simultaneously.

We clarified this point on lines 327–330 in revised Methods.

Methods: alpha is used twice (learning rate and dt/tau)

Reply:

We resolved the ambiguity in notation, we changed to $\gamma = dt/\tau$, leaving learning rate to be denoted as α .

This change is reflected on lines 333–334 in revised Methods.

(Remarks on code availability):

I reviewed the code just very briefly. I couldn't easily find where the code for training RNNs is (maybe in the yaml files?). I am not sure if it has not been shared, and only the RNN data is shared, or if I couldn't find it.

Reply:

We provide links in the Code Availability section to all three packages: one for training RNNs, one for analyzing neural representations of RNNs, and one for fitting latent circuits.

We now added the links to the code on lines 338–339, 506–507, and 479–480 in the corresponding revised Methods sections.

Reviewer #2 (Remarks to the Author):

This study is a carefully considered, important analysis of the biases that different activation functions endow RNN solutions to various tasks. Rather than just highlighting the differences, this study also tries to identify why different activation functions have such different solutions. The latent circuit inference and the evaluation of how the networks handle out-of-distribution generalization provided valuable insights into their underlying mechanisms. The findings from both analyses underscored the significance of this study, highlighting the critical role of activation functions in network design and the potential risks of overlooking them. The inclusion of networks with and without Dale's Law was a good addition to the paper and provided a good benchmark for comparison of the size of the effects as noted by the authors at the end of the results section. The discussion was excellent. Given the biological motivation of the paper, I particularly appreciated the extensive comparison to experimental data and consideration of what may be biologically plausible.

Reply:

We thank the reviewer for recognizing the broad significance of our work and for thoughtful comments that helped us improve the manuscript.

Minor points:

- The manuscript states that the top 30 RNNs from each network architecture were selected for analysis. Providing insight into the performance of the remaining 70 networks would strengthen the paper and clarify the significance of this cutoff in shaping the results.

Reply:

We originally restricted our analyses to the top 30 RNNs from each architecture because computing MDS embedding of all 1,200 networks (100×3 [ReLU / sigmoid / tanh] $\times 2$ [Dale / no Dale] $\times 2$ [trained / shuffled]) is computationally costly. Therefore, we restricted the analysis in the original manuscript to 360 networks (30×3 [ReLU / sigmoid / tanh] $\times 2$ [Dale / no Dale] $\times 2$ [trained / shuffled]) to reduce computational cost. Prompted by the reviewer's comment, we extended all analyses presented in the main text to include the top 50 networks from each architecture (Fig. 2, Fig. 3, Fig. 5). In addition, we performed the MDS embedding on the remaining 50 networks (Extended Data Fig. 2). The results for the top 50 and the remaining 50 networks were qualitatively similar. We further verified that all networks were trained to a similar level of performance, with each achieving the coefficient of determination $R^2 > 0.84$ between RNN output and targets on test data (Extended Data Fig. 1, Extended Data Table 1).

We present these analyses in revised Figs. 2,3,5, in new Extended Data Figs. 1,2, Extended Data Table 1 and on lines 40–43 in revised Results.

- There is a brief mention of implications for RNNs fit to reproduce neural data. Whilst I understand it is outside of the scope of this work to directly examine the effect of different activation functions on these methods, a little more detail here would be beneficial to the discussion.

Reply:

Our findings suggest that some architectures may be inherently more suitable for modeling biological neural activity. While RNNs can approximate any dynamics given a sufficient number of units, the number of units required to achieve a given level of accuracy depends on how well the architecture aligns with the structure and constraints of biological circuits. Just like functions can be more compactly represented using an appropriate set of basis functions, certain architectures may capture neural data more efficiently by requiring fewer units. Architectures more closely aligned with that of biological circuits are thus expected to require fewer units to fit the neural dynamics accurately.

Moreover, it remains an open question whether two architectures that fit the same neural responses equally well will arrive at the same underlying circuit solution.

We have addressed this point on lines 206–215 in revised Discussion.

- Some more thought might be given in either the discussion or the results as to other activation functions and whether each activation function causes unique network behavior or rather there are different groups (such as the similarities highlighted between ReLU and sigmoid networks in this paper).

Reply:

We thank the reviewer for raising this interesting point. In the CDDM task, our results indicate that ReLU and sigmoid networks produced solutions that were more similar to each other than to the solutions in tanh RNNs. In contrast, in the Go/NoGo and Memory Number tasks, sigmoid networks often produced solutions that were as distinct from ReLU networks as from tanh RNNs. These observations suggest that similarities between activation functions may be task-dependent rather than absolute. For some tasks, the positivity of unit activations may play a critical role, leading ReLU and sigmoid networks to converge on similar solutions. For other tasks, saturation properties may be more important, resulting in greater similarity between tanh and sigmoid networks than between either and ReLU RNNs. As another consideration, it is difficult to envision a task in which differences between ReLU and softplus activations $\frac{1}{\beta} \log(1 + e^{\beta x})$ with large β would significantly affect the solutions, as softplus approaches the ReLU function asymptotically when β becomes large. Yet, decreasing the parameter β renders the softplus activation progressively more linear, causing it to deviate significantly from ReLU. When the activation function becomes sufficiently linear for small β , the network may gradually—or even abruptly—lose its ability to solve complex nonlinear tasks altogether. Thus, whether activation functions produce similar or distinct circuit solutions may depend on the specific computation, and even small changes in the activation function can, in principle, lead to discontinuous shifts in the resulting solution.

We clarified this point on lines 238–250 in revised Discussion.

Reviewer #2 (Remarks on code availability):

I have not run the code but I have looked through the repository and it seems well documented with a decent README.

Reply:

We appreciate the review of our code.

Reviewer #1 (Remarks to the Author):

The authors have addressed in details all my concerns with the previous version of the manuscript. Mainly, the Methods are now complete and it has been clarified what part of the effects shown in the study can be attributed to learning, and which effects correspond to differences present in untrained networks.

I believe this is a very useful study for the community. It analyzes in detail how different structural properties can drastically change the emergent properties in recurrent networks.

Reply:

We thank the reviewer again for thoughtful comments, which helped to strengthen our results and significantly improve the quality of the manuscript.

Reviewer #2 (Remarks to the Author):

Thank you for addressing some of my points in the discussion and extending the analyses to the top 50 networks and the MDS to all the networks.

Thank you for the detailed explanation clarifying that similarities between activation functions appear to be task-dependent, with predictable relationships arising from the interaction between activation properties and specific computational demands.

Reply:

We are grateful to the reviewer once again for the invaluable feedback, which has significantly strengthened the quality and clarity of this manuscript.

As a suggestion for strengthening the manuscript, it might be valuable to consider whether there are systematic ways to determine, from task properties themselves, whether a task requires strongly positive-only activations (e.g., benefits from positivity constraints) or relies on saturation effects. Identifying measurable features or computational demands of a task that predict when positivity or saturation properties become critical could help establish more general principles linking task characteristics to activation-function similarities—and clarify how one could know ahead of time which tasks are likely to depend on specific activation properties. I think this would be valuable for anyone inspired by this thought-provoking work to change their approach to this type of modelling.

Reply:

We thank the reviewer for this forward-looking suggestion. Establishing systematic links between task requirements and specific properties of activation functions would indeed be highly valuable, both for theory and for guiding model design.

At present, there is no formal framework that predicts which activation function is optimal for a given set of task demands. Nevertheless, an instructive analogy comes from function approximation: certain basis functions are inherently better suited to representing specific classes of functions, leading to more compact representations. For example, periodic functions are efficiently represented by Fourier bases. Similarly, the choice of activation function in neural networks can interact with task structure so that particular activation properties enable more efficient solutions.

As an example, consider the 3-bit flip-flop task, in which three outputs store independent memory bits, each set by transient +1 or -1 pulses from corresponding inputs and held until the next pulse (Sussillo & Barak, *Neural Comput* 2013). Tanh RNNs can solve this task with just three units, as a single tanh unit can sustain bistable activity—positive

or negative—through strong self-excitation. In contrast, a single ReLU unit cannot produce bistability, requiring a larger recurrent circuit to maintain and flip each bit in ReLU RNNs.

Overall, a formal theory of which activation functions suit specific task demands is lacking, and developing systematic principles remains an important direction for future research.

We addressed this point in the sixth paragraph of revised Discussion.

Reviewer #2 (Remarks on code availability):

I have not run the code but I have looked through the repository and it seems well documented with a decent README.

Reply:

We appreciate the review of our repository.